# SCOPE: Saliency-Coverage Oriented Token Pruning for Efficient Multimodel LLMs

**Jinhong Deng[1,3,4], Wen Li[2],[*] Joey Tianyi Zhou[3,4], Yang He[3,4]**
[1]University of Electronic Science and Technology of China (UESTC)
[2]Shenzhen Institute for Advanced Study, UESTC
[3]CFAR, Agency for Science, Technology and Research (A*STAR), Singapore
[4]IHPC, Agency for Science, Technology and Research (A*STAR), Singapore
{jhdengvision, liwenbnu}@gmail.com,
{joey_zhou, he_yang}@a-star.edu.sg

## Abstract

Multimodal Large Language Models (MLLMs) typically process a large number of visual tokens, leading to considerable computational overhead, even though many of these tokens are redundant. Existing visual token pruning methods primarily focus on selecting the most salient tokens based on attention scores, resulting in the semantic incompleteness of the selected tokens. In this paper, we propose a novel visual token pruning strategy, called **S**aliency-**C**overage **O**riented token **P**runing for **E**fficient MLLMs (SCOPE), to jointly model both the saliency and coverage of the selected visual tokens to better preserve semantic completeness. Specifically, we introduce a set-coverage for a given set of selected tokens, computed based on the token relationships. We then define a token-coverage gain for each unselected token, quantifying how much additional coverage would be obtained by including it. By integrating the saliency score into the token-coverage gain, we propose our SCOPE score and iteratively select the token with the highest SCOPE score. We conduct extensive experiments on multiple vision-language understanding benchmarks using the LLaVA-1.5 and LLaVA-Next models. Experimental results demonstrate that our method consistently outperforms prior approaches. Our code is available at https://github.com/kinredon/SCOPE.

## 1 Introduction

Recent advances in Multimodal Large Language Models (MLLMs)[24, 25, 51, 20, 21] have significantly advanced open-ended visual understanding tasks[12, 27, 47, 8] by integrating powerful vision encoders [34] with autoregressive large language models [37, 1]. These systems typically tokenize visual inputs into sequences of patch-level embeddings (*i.e.*, visual tokens), which are then fed into the language model via either projection modules [24] or attention-based fusion mechanisms [19]. Despite its effectiveness, this paradigm incurs substantial computational overhead, particularly when processing high-resolution images or temporally dense video inputs. For instance, a ViT encoder [11] applied to a $448 \times 448$ image can generate over 1,000 visual tokens. This number increases rapidly in high-resolution and video scenarios involving multiple frames. Since these tokens are jointly processed with textual tokens, the computational cost of self-attention grows quadratically with the number of visual tokens [30, 25], limiting their deployment in practical applications such as edge computing and robotics [17, 33, 44].

---

[*]Corresponding Author.
This work was completed during Jinhong Deng's internship at CFAR, A*STAR.

39th Conference on Neural Information Processing Systems (NeurIPS 2025).

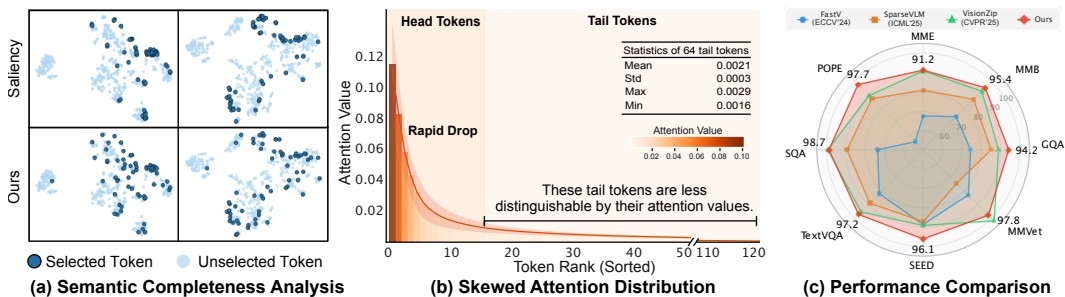

Figure 1: (a) Semantic Completeness Analysis. We visualize the selected tokens using a saliency-based rule (Top) and our method (Bottom). The saliency score corresponds to the visual attention assigned to the CLS token. Our method selects tokens that maximize coverage while preserving the most dominant visual information. (b) Skewed Attention Distribution. We show the averaged attention distribution of the top 128 tokens on the MME benchmark. The attention weights rapidly flatten, making tail tokens less distinguishable based on their attention values. (c) Performance comparison with prior methods across various benchmarks. The model is LLaVA-1.5 7B, and the number of retained tokens is 64.

However, not all visual tokens contribute equally to the final outputs of the language model [7]. Many background or repetitive patches carry redundant or less informative content [6, 11]. This motivates the need for efficient visual token pruning or compression, aiming to retain only the most relevant tokens while discarding those that are redundant. To this end, recent works [7, 41, 49] have proposed various pruning strategies that select salient visual tokens based on attention scores, *i.e.*, visual attention from text prompts or from the CLS token in vision transformers. For instance, VisionZIP [43] selects visual tokens that receive the highest attention from the CLS token.

While effective, saliency-based visual token pruning methods exhibit notable limitations in complex vision-language tasks. First, they inevitably **compromise semantic completeness** by discarding key contextual information essential for comprehensive visual understanding. For example, in response to the question "Where is the cat?", attention may focus primarily on the object "cat" while neglecting its surrounding context. The saliency-based methods typically concentrate on a small subset of visual tokens (see Fig. 1(a)), resulting in significant semantic loss. Moreover, saliency-based approaches often suffer from highly **skewed attention distribution**, where only a few tokens receive substantial attention while the rest exhibit nearly uniform (*i.e.*, flat) attention values as shown in Fig. 1(b). This hampers the discriminability among tokens, making it difficult to differentiate potentially informative ones from truly redundant ones.

To address the above challenges, we propose a novel visual token pruning strategy, named **S**aliency-**C**overage **O**riented token **P**runing for **E**fficient MLLMs (SCOPE), which jointly models the saliency and coverage of selected visual tokens to preserve semantic completeness. Specifically, we first define a set-coverage score for a selected token set based on token relationships and introduce a token-coverage gain for each unselected token, measuring the additional coverage achieved by including that token. We then propose a SCOPE score to integrate the token saliency score into the token-coverage gain, and iteratively select the token with the highest SCOPE score. This enables our method to retain tokens that not only contribute the most salient information but also ensure broad semantic coverage (see Fig. 1(a)).

To evaluate the effectiveness of our SCOPE, we conduct extensive experiments on a variety of vision-language understanding benchmarks using popular MLLMs, including LLaVA-1.5 [24] and LLaVA-Next [25]. The results demonstrate that our method consistently outperforms prior approaches by a significant margin (see Fig. 1(c)). For instance, SCOPE achieves a 9× reduction in the number of visual tokens while retaining 96.0% of the original performance on LLaVA-1.5 7B [24].

Our main contributions are summarized as follows:

- We reveal the limitation of the saliency-based visual token pruning methods, which unfortunately ignore the semantic completeness of the selected visual tokens and suffer from a highly skewed attention distribution problem.

- We propose a novel visual token pruning strategy, named **S**aliency-**C**overage **O**riented token **P**runing for **E**fficient MLLMs (SCOPE), which jointly models saliency and coverage of the retained visual tokens to preserve semantic completeness.
- We integrate SCOPE into representative MLLMs such as LLaVA-1.5 and LLaVA-Next without training, and demonstrate its effectiveness on multiple vision-language benchmarks, achieving a favorable trade-off between computational efficiency and task performance.

## 2   Related Work

**Multimodal Large Language Models (MLLMs).** Large Language Models (LLMs)[1, 37, 3, 10, 16] have achieved remarkable success in a wide range of language understanding and generation tasks. Building on this foundation, Multimodal LLMs (MLLMs)[24, 25, 21, 50, 19, 4] have shown impressive progress in visual understanding. A prevailing paradigm in MLLMs projects visual features into a sequence of visual tokens via a vision-to-language projector, and feeds them into the LLM alongside text tokens, as exemplified by LLaVA [24, 25], Qwen-VL [4], and Mini-Gemini [21].

However, real-world images are often high-resolution, resulting in long visual token sequences that significantly slow down inference in MLLMs [23, 30, 20, 9]. For example, LLaVA-Next [25] converts a $672 \times 672$ image into over 2,000 tokens. The situation worsens when handling multiple images or videos, further increasing the number of visual tokens. This highlights the need for effective strategies to reduce token length and accelerate vision-language inference.

**Visual Token Pruning/Compression in MLLMs.** A number of recent studies [49, 41, 40, 7] have focused on reducing visual token redundancy in MLLMs without requiring additional model training. Most of these methods [7, 49, 41] rely on specific attention scores to rank token saliency, such as text-to-vision attention in LLMs or CLS-token attention in vision transformers. They typically retain only the top-ranked tokens using a top-$k$ strategy, *i.e.*, selecting tokens with the highest attention scores. For instance, FastV [7] leverages early-layer text-to-vision attention to retain salient tokens. SparseVLM [49] uses important textual words as a rater to guide token selection. VisionZip [43] applies CLS-based attention in the vision transformer for token pruning. To further increase the information density of the selected tokens, several approaches attempt to merge semantically similar tokens [43, 49, 35]. DivPrune [2] selects visual tokens by maximizing the diversity of selected tokens. In contrast, our method jointly considers both saliency and coverage, aiming to preserve semantic completeness while reducing token redundancy.

## 3   Method

In this section, we first introduce the preliminaries of visual token pruning and discuss the instantiation of saliency-based pruning methods in Sec. 3.1. In Sec.3.2, we provide a coverage analysis and show that saliency-based methods often suffer from low coverage. Finally, we present our proposed **S**aliency-**C**overage **O**riented token **P**runing for **E**fficient MLLMs (SCOPE) in Sec.3.3.

### 3.1   Preliminary

**Visual Token Pruning.** The core architecture of LLMs consists of stacked self-attention layers and feed-forward networks (FFNs)[38], where the computational complexity grows quadratically with the input sequence length. In MLLMs, input images are typically high-resolution, resulting in long sequences of visual tokens. For instance, LLaVA[26] produces 576 visual tokens for a single image, which is often significantly longer than the corresponding text input in many visual understanding tasks. Furthermore, visual tokens often exhibit substantial redundancy [7, 41] due to repeated patterns and limited informational content in background regions.

Therefore, reducing the number of visual tokens is essential for enhancing the computational efficiency of MLLMs. In particular, $\mathcal{V} = \{v_1, \ldots, v_N\}$ denotes the full set of $N$ visual tokens extracted from the image, where each token $v_i \in \mathbb{R}^d$ represents a local region of the image. The goal of visual token pruning algorithm $\mathcal{A}$ is to select a small subset of visual tokens $\mathcal{S} = \{v_1, \ldots, v_K\} = \mathcal{A}(\mathcal{V})$, where $K \ll N$. The objective of visual token pruning is to ensure that the model's output based on $\mathcal{S}$ closely approximates the output based on the full set $\mathcal{V}$. Formally, the pruning objective can be

formulated as:

$$\min_{\mathcal{S}} \ \mathcal{L}\left(\mathcal{M}(\mathcal{S}, T), \ \mathcal{M}(\mathcal{V}, T)\right), \tag{1}$$

where $\mathcal{M}(\cdot, T)$ denotes the output of the vision-language model given visual input (either $\mathcal{V}$ or $\mathcal{S}$) and text input $T$, and $\mathcal{L}$ is a function to measure the output difference of LLM.

**Saliency-based Visual Token Pruning.** The saliency-based visual token pruning methods aim to reduce token redundancy by retaining the most salient visual tokens while discarding the less informative ones. The core challenge lies in how to effectively measure the saliency of each visual token. Several prior works [7, 41, 49, 43] estimate saliency by leveraging attention scores. Specifically, the attention matrix $\boldsymbol{A}$ is calculated as:

$$\boldsymbol{A} = \text{Softmax}\left(\frac{\boldsymbol{Q}\boldsymbol{K}^\top}{\sqrt{d}}\right), \tag{2}$$

where $d$ is the embedding dimension, $\boldsymbol{Q}$ and $\boldsymbol{K}$ is the query and key matrices in the standard attention mechanism. These attention scores indicate the interaction strength between tokens, guiding the identification of highly salient tokens. In practice, in the vision encoder of CLIP [34], the [CLS] token is used to aggregate global information from the entire image. Therefore, the attention scores from the [CLS] token to the visual tokens serve as a reasonable proxy for token saliency. Based on these saliency scores, token pruning methods typically adopt a top-k selection strategy to retain only the most salient visual tokens. This approach effectively reduces visual token redundancy and significantly accelerates MLLM inference across various tasks.

## 3.2 Coverage Analysis

Although saliency-based pruning methods can effectively identify important tokens based on attention scores, they inevitably discard certain semantically critical tokens that are essential for comprehensive visual understanding. Semantic completeness, however, is crucial for accurately responding to a wide range of instruction prompts in MLLMs. Furthermore, saliency-based approaches often suffer from highly skewed attention distributions, where a small subset of tokens receives disproportionately high attention, while the remaining tokens exhibit nearly uniform (i.e., flat) attention values. This skewness undermines token discriminability, making it challenging to distinguish between potentially informative tokens and truly redundant ones. To quantitatively assess the representational completeness of the selected

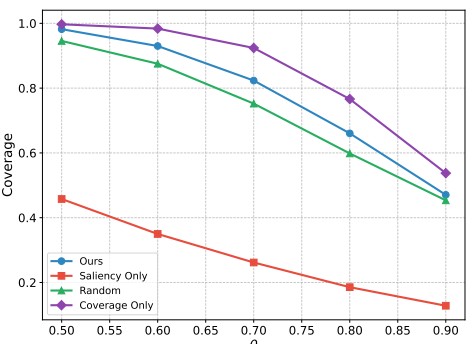

Figure 2: Comparison of $\theta$-coverage across different token pruning criteria. The experiments are conducted on the MME benchmark, with 64 tokens selected out of the original 576 in LLaVA 1.5 7B.

tokens, we introduce the notion of the $\theta$-coverage (see **Definition** 1), which measures the degree to which the retained tokens cover the semantic space of the full token set.

**Definition 1** ($\theta$-Coverage). *Let $\mathcal{V} = \{v_i \in \mathbb{R}^d \mid i = 1, ..., n\}$ denote the full set of tokens extracted from an input image, and let $\mathcal{V}' \subseteq \mathcal{V}$ be a subset of selected tokens. For a given similarity threshold $\theta \in [0, 1]$, we say that a token $v \in \mathcal{V}$ is* covered *by $\mathcal{V}'$ if there exists at least one token $v' \in \mathcal{V}'$ such that their cosine similarity satisfies:*

$$\text{sim}(v, v') := \frac{v^\top v'}{\|v\| \cdot \|v'\|} \geq \theta. \tag{3}$$

*The $\theta$-coverage of $\mathcal{V}'$ over $\mathcal{V}$ is then defined as the proportion of tokens in $\mathcal{V}$ that are covered by $\mathcal{V}'$:*

$$\text{Coverage}_\theta(\mathcal{V}', \mathcal{V}) = \frac{1}{|\mathcal{V}|} \sum_{v \in \mathcal{V}} \mathbb{I}\left(\max_{v' \in \mathcal{V}'} \text{sim}(v, v') \geq \theta\right), \tag{4}$$

*where $\mathbb{I}(\cdot)$ is the indicator function, which equals 1 if the condition holds and 0 otherwise.*

*This definition provides a semantic-aware metric to quantify how well the selected tokens set $\mathcal{V}'$ represents the full set. A higher value of $\theta$ imposes a stricter similarity criterion, typically leading to lower coverage but ensuring that the retained tokens are more semantically representative.*

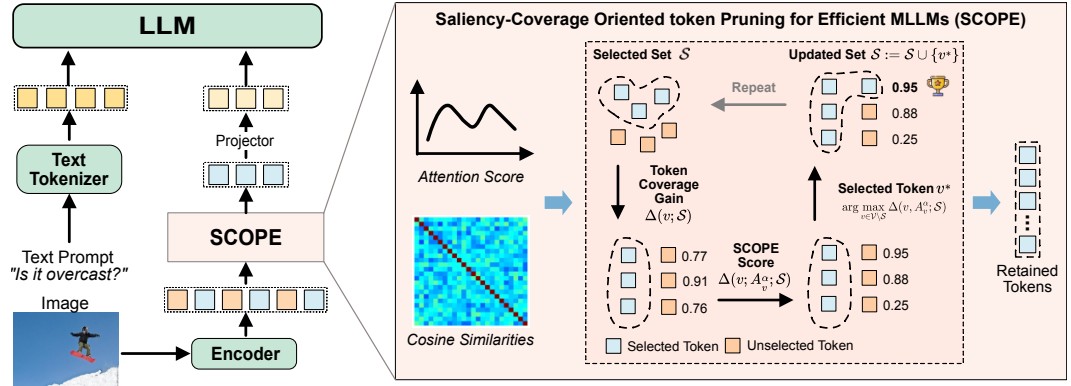

Figure 3: An overview of the proposed visual token pruning framework. The left part illustrates how our method reduces the number of visual tokens before feeding them into the LLM, thereby accelerating inference in MLLMs without requiring additional model training. The right part provides a detailed view of our SCOPE method, which jointly optimizes saliency and coverage to select a compact yet semantically representative subset of visual tokens.

In particular, we present the $\theta$-coverage results on the MME benchmark in Fig. 2. The Saliency Only method selects dominant tokens solely based on the attention scores from the `CLS` token. However, it consistently exhibits low coverage across different values of $\theta$, even performing worse than the random selection baseline. This observation suggests that although the saliency-based method captures dominant information, it tends to overlook a substantial amount of semantic content. In contrast, our method (detailed in Sec.3.3) incorporates saliency scores into a coverage-aware selection framework, striking a better balance between saliency and semantic coverage. As a result, it achieves significantly higher coverage compared to the Saliency Only method.

## 3.3 Saliency-Coverage Oriented Token Pruning

In contrast to saliency-based pruning methods, our goal is to jointly optimize saliency and coverage in the visual token selection process. This enables the pruning algorithm to not only preserve the most informative tokens but also maximize the semantic coverage of the selected subset. As a result, the retained tokens are both highly informative and semantically diverse, thereby maintaining semantic completeness under a constrained token budget, which is an essential property for comprehensive visual understanding across a wide range of multimodal tasks.

In the following, we first define the notion of coverage for selected tokens. Next, we introduce the concept of token-coverage gain, *i.e.*, the additional coverage obtained by including a new token in the selected set [14]. Finally, we incorporate the saliency score into the token-coverage gain formulation to balance both selection criteria. The overview of the proposed method is presented in Fig. 3.

**Set-coverage for selected tokens.** To quantify semantic coverage, we measure the similarity between token vectors using cosine similarity. We first define the individual coverage score $C(u, \mathcal{S})$ for a token $u \in \mathcal{V}$ by a set of selected tokens $\mathcal{S} \subseteq \mathcal{V}$ as:

$$C(u, \mathcal{S}) = \max_{s \in \mathcal{S}} \text{sim}(u, s) \tag{5}$$

where $\text{sim}(u, s)$ is the cosine similarity metric between token $u$ and token $s$. The overall coverage of the selected subset $\mathcal{S}$ is defined as the sum of the maximum similarities between each token in the full set $\mathcal{V}$ and its most similar token in $\mathcal{S}$:

$$f(\mathcal{S}) = \sum_{u \in \mathcal{V}} C(u, \mathcal{S}) = \sum_{u \in \mathcal{V}} \max_{s \in \mathcal{S}} \text{sim}(u, s) \tag{6}$$

This formulation encourages the selection of tokens that are semantically diverse and broadly representative of the input space. Intuitively, it ensures that each token in the full set has at least one similar counterpart in the selected subset, thus preserving information while reducing the token count.

**Token-coverage Gain.** To quantify the contribution of each candidate token $v \in \mathcal{V} \setminus \mathcal{S}$, we evaluate its *marginal gain* with respect to the current subset $\mathcal{S}$ [14]. The marginal gain is defined as the

**Algorithm 1** SCOPE

---

**Require:** A full set of tokens $\mathcal{V} = \{v_1, ..., v_n\} \subset \mathbb{R}^d$, number of retained token $K$, pairwise
  similarities $S_{uv} = \text{sim}(u, v)$ for all $u, v \in \mathcal{V}$, attention score $A_v$ for each token $v$, and a scaling
  factor $\alpha$.
**Ensure:**
  Selected token subset $\mathcal{S} \subseteq \mathcal{V}$ with $|\mathcal{S}| = K$
 1: Initialize $\mathcal{S} \leftarrow \emptyset$                                                                     ▷ Start with an empty selected subset
 2: Initialize coverage scores: $c_u \leftarrow 0$ for all $u \in \mathcal{V}$    ▷ $c_u$ tracks the best similarity between $u$ and
  any selected token so far
 3: **for** $t = 1$ to $K$ **do**
 4:   **for all** $v \in \mathcal{V} \setminus \mathcal{S}$ **do**
 5:     Compute marginal gain: $\Delta(v; \mathcal{S}) = \sum_{u \in \mathcal{V}} [\max(S_{uv}, c_u) - c_u]$         ▷ Compute the
  additional coverage that token $v$ brings if added to $\mathcal{S}$
 6:   **end for**
 7:   Select next token: $v^* \in \arg\max_{v \in \mathcal{V} \setminus \mathcal{S}} \Delta(v; \mathcal{S}) \cdot A_v^\alpha$      ▷ Choose the token that balances
  coverage and saliency score
 8:   Update selected subset: $\mathcal{S} \leftarrow \mathcal{S} \cup \{v^*\}$                                  ▷ Add the selected token to the subset
 9:   Update coverage scores: $c_u \leftarrow \max(c_u, S_{uv^*})$   $\forall u \in \mathcal{V}$ ▷ Update coverage scores using the
  newly added token
10: **end for**
11: **return** $\mathcal{S}$

---

increase in total coverage achieved by including $v$, and can be formally expressed as follows:

$$\Delta(v; \mathcal{S}) = f(\mathcal{S} \cup \{v\}) - f(\mathcal{S}), \tag{7}$$

Expanding this definition using Eq. (6), we can express the marginal gain as the sum of the new
coverage provided by $v$ to each token $u$ that was not already fully covered by $\mathcal{S}$:

$$\Delta(v; \mathcal{S}) = \sum_{u \in \mathcal{V}} \max_{s \in (\mathcal{S} \cup \{v\})} \text{sim}(u, s) - \sum_{u \in \mathcal{V}} \max_{s \in \mathcal{S}} \text{sim}(u, s)$$
$$= \sum_{u \in \mathcal{V}} (\max(C(u, \mathcal{S}), \text{sim}(u, v)) - C(u, \mathcal{S})) \tag{8}$$

This quantifies how much additional coverage is achieved by selecting token $v$, taking into account
its ability to represent other tokens $u \in \mathcal{V}$ that are not yet well-represented by the current subset $\mathcal{S}$.

**SCOPE score.** While the token-coverage gain considers only the geometric coverage in semantic
space, it overlooks the intrinsic information carried by individual tokens. To address this limitation,
we propose the SCOPE gain, which incorporates token saliency into the coverage gain to better
preserve visual token information. Specifically, we integrate the visual attention score into the
coverage gain function as follows:

$$\Delta(v, A_v^\alpha; \mathcal{S}) = \Delta(v; \mathcal{S}) \cdot A_v^\alpha, \tag{9}$$

where $A_v^\alpha$ denotes the attention score of the visual token $v$, and $\alpha$ is a scaling factor. The token $v^*$
with the highest SCOPE gain is selected and added to the subset $\mathcal{S}$:

$$v^* \in \arg\max_{v \in \mathcal{V} \setminus \mathcal{S}} \Delta(v, A_v^\alpha; \mathcal{S}) \tag{10}$$

This process is iteratively repeated until the desired subset size is reached. The pseudocode of the
proposed pruning method is presented in Algorithm 1.

**Integration into MLLMs.** The proposed method is applicable to a wide range of MLLMs. In
this work, we apply it to the widely adopted LLaVA[26] and LLaVA-Next [25] models, following
prior studies [7, 49, 41]. Our method is integrated after the vision encoder to maximize information
retention post token pruning. This enables the language model to receive more complete visual signals,
thereby supporting comprehensive visual understanding without compromising performance. Our
method is train-free and significantly accelerates the inference of MLLMs with minimal performance
degradation. For example, our approach preserves over 96% of the original model's performance
while reducing the number of visual tokens by a factor of 8 in LLaVA 1.5 7B.

Table 1: **Performance comparison under different vision token configurations.** We evaluate the LLaVA 1.5 7B model, where the default number of visual tokens is 576. The first row for each method reports the raw accuracy across benchmarks, and the second row indicates the performance relative to the upper bound. $^{\dagger}$ denotes the results adapted from [49].

| Method | GQA | MMB | MME | POPE | SQA | TextVQA | SEED | MMVet | Avg. |
|---|---|---|---|---|---|---|---|---|---|
| Upper Bound, 576 Tokens (100%) | | | | | | | | | |
| Vanilla (CVPR'24) | 61.9 | 64.7 | 1862 | 85.9 | 69.5 | 58.2 | 58.6 | 31.1 | 100% |
| | 100% | 100% | 100% | 100% | 100% | 100% | 100% | 100% | |
| Retain 192 Tokens (↓ 66.7%) | | | | | | | | | |
| FastV (ECCV'24) | 52.7 | 61.2 | 1612 | 64.8 | 67.3 | 52.5 | 57.1 | 27.7 | 89.5% |
| | 85.1% | 94.6% | 86.6% | 75.4% | 96.8% | 90.2% | 97.4% | 89.7% | |
| SparseVLM (ICML'25) | 57.6 | 62.5 | 1721 | 83.6 | 69.1 | 56.1 | 55.8 | 31.5 | 96.5% |
| | 93.1% | 96.6% | 92.4% | 97.3% | 99.4% | 96.4% | 95.2% | 101.3% | |
| VisionZip (CVPR'25) | 59.3 | 63.0 | 1783 | 85.3 | 68.9 | 57.3 | 56.4 | 31.7 | 98.0% |
| | 95.8% | 97.4% | 95.7% | 99.3% | 99.1% | 98.5% | 96.2% | 101.9% | |
| PDrop (CVPR'25)$^{\dagger}$ | 57.1 | 63.2 | 1766 | 82.3 | 70.2 | 56.1 | 54.7 | 30.5 | 96.2% |
| | 92.2% | 97.7% | 94.8% | 95.8% | 101.0% | 96.4% | 93.3% | 98.1% | |
| Ours | 60.1 | 63.6 | 1804 | 86.4 | 68.8 | 57.7 | 58.7 | 32.5 | 99.5% (↓ 0.5%) |
| | 97.1% | 98.3% | 96.9% | 100.6% | 99.0% | 99.1% | 100.2% | 104.5% | |
| Retain 128 Tokens (↓ 77.8%) | | | | | | | | | |
| FastV (ECCV'24) | 49.6 | 56.1 | 1490 | 59.6 | 60.2 | 50.6 | 55.9 | 28.1 | 84.4% |
| | 80.1% | 86.7% | 80.0% | 69.4% | 86.6% | 86.9% | 95.4% | 90.4% | |
| SparseVLM (ICML'25) | 56.0 | 60.0 | 1696 | 80.5 | 67.1 | 54.9 | 53.4 | 30.0 | 93.3% |
| | 90.5% | 92.7% | 91.1% | 93.7% | 96.5% | 94.3% | 91.1% | 96.5% | |
| VisionZip (CVPR'25) | 57.6 | 62.0 | 1761.7 | 83.2 | 68.9 | 56.8 | 54.9 | 32.6 | 96.9% |
| | 93.1% | 95.8% | 94.6% | 96.9% | 99.1% | 97.6% | 93.7% | 104.8% | |
| PDrop (CVPR'25)$^{\dagger}$ | 56 | 61.1 | 1664 | 82.3 | 69.9 | 55.1 | 53.3 | 30.8 | 94.4% |
| | 90.5% | 94.4% | 89.4% | 95.8% | 100.6% | 94.7% | 91.0% | 99.0% | |
| Ours | 59.7 | 62.5 | 1776 | 86.1 | 68.4 | 57.2 | 57.8 | 31.4 | 98.1% (↓ 1.9%) |
| | 96.4% | 96.6% | 95.4% | 100.2% | 98.4% | 98.3% | 98.6% | 101.0% | |
| Retain 64 Tokens (↓ 88.9%) | | | | | | | | | |
| FastV (ECCV'24) | 46.1 | 48.0 | 1256 | 48 | 51.1 | 47.8 | 51.9 | 25.8 | 74.9% |
| | 74.5% | 74.2% | 67.5% | 55.9% | 73.5% | 82.1% | 88.6% | 83.0% | |
| SparseVLM (ICML'25) | 52.7 | 56.2 | 1505 | 75.1 | 62.2 | 51.8 | 51.1 | 23.3 | 85.1% |
| | 85.1% | 86.9% | 80.8% | 87.4% | 89.5% | 89.0% | 87.2% | 74.9% | |
| VisionZip (CVPR'25) | 55.1 | 60.1 | 1690 | 77.0 | 69.0 | 55.5 | 52.2 | 31.7 | 93.5% |
| | 89.0% | 92.9% | 90.8% | 89.6% | 99.3% | 95.4% | 89.1% | 101.9% | |
| PDrop (CVPR'25)$^{\dagger}$ | 41.9 | 33.3 | 1092 | 55.9 | 69.2 | 45.9 | 40.0 | 30.7 | 73.5% |
| | 67.7% | 51.5% | 58.6% | 65.1% | 99.6% | 78.9% | 68.3% | 98.7% | |
| Ours | 58.3 | 61.7 | 1698 | 83.9 | 68.6 | 56.6 | 56.3 | 30.4 | 96.0% (↓ 4.0%) |
| | 94.2% | 95.4% | 91.2% | 97.7% | 98.7% | 97.3% | 96.1% | 97.7% | |

# 4 Experiment

## 4.1 Experiments Setup

**Evaluation Benchmarks and Baselines.** Following prior work[49], we evaluate the effectiveness of the proposed method using a set of widely adopted multimodal benchmarks. Specifically, these include GQA [13], MMBench[27], POPE [22], ScienceQA[29], TextVQA [36], SEEDBench[18], and MMVet [45]. We also compare against several state-of-the-art baselines, including FastV[7], SparseVLM [49], VisionZip[43], and PDrop [41]. For the video benchmarks, we evaluate the MLLMs on the benchmarks TGIF [15], MSVD [5], MSRVTT [42], and ActivityNet [46]. For further details on evaluation benchmarks and metrics, we refer the reader to the Appendix B.

**Implementation Details.** We integrate the proposed method into LLaVA 1.5 [26] and LLaVA-Next [25] for image understanding and Video-LLaVA [23] for video understanding. The pruning module is inserted after the vision encoder. The saliency score is derived from the attention weights of visual tokens with respect to the CLS token at the second-to-last layer (layer -2) of the vision encoder. The scaling factor $\alpha$ is set to 1.0 by default. Our implementation is based on the lmms-evals [48] package. We conduct the experiments on $4 \times A100$ GPUs. The inference batch size is set to 1 for all the evaluation results.

Table 2: **Performance comparison under different vision token configurations.** The evaluated model is LLaVA-Next 7B. The vanilla number of vision tokens is 2,880. The first line of each method is the raw accuracy of benchmarks, and the second line is the proportion relative to the upper bound.

| Method | GQA | MMB | MME | SQA | TextVQA | MMMU | Avg. |
|--------|-----|-----|-----|-----|---------|------|------|
| Upper Bound, 2880 Tokens (100%) | | | | | | | |
| Vanilla (CVPR'24) | 64.2 | 67.9 | 1842 | 70.2 | 61.3 | 35.1 | 100% |
| | 100% | 100% | 100% | 100% | 100% | 100% | |
| Retain 640 Tokens (↓ **77.8%**) | | | | | | | |
| SparseVLM (ICML'25) | 60.3 | 65.7 | 1772 | 67.7 | 57.8 | 34.6 | 96.0% |
| | 93.9% | 96.8% | 96.2% | 96.4% | 94.3% | 98.6% | |
| VisionZip (CVPR'25) | 61.3 | 66.3 | 1787 | 68.1 | 60.2 | 34.7 | 97.4% |
| | 95.5% | 97.6% | 97.0% | 97.0% | 98.2% | 98.9% | |
| Ours | 61.9 | 66.2 | 1842 | 67.8 | 60.1 | 36.9 | 98.9% (↓ **1.1%**) |
| | 96.4% | 97.5% | 100.0% | 96.6% | 98.0% | 105.1% | |
| Retain 320 Tokens (↓ **88.9%**) | | | | | | | |
| SparseVLM (ICML'25) | 57.7 | 64.3 | 1694 | 67.3 | 55.9 | 34.4 | 93.6% |
| | 89.9% | 94.7% | 92.0% | 95.9% | 91.2% | 98.0% | |
| VisionZip (CVPR'25) | 59.3 | 63.1 | 1702 | 67.3 | 58.9 | 35.3 | 95.0% |
| | 92.4% | 92.9% | 92.4% | 95.9% | 96.1% | 100.6% | |
| Ours | 61.0 | 65.9 | 1789 | 67.7 | 58.4 | 35.6 | 97.1% (↓ **2.9%**) |
| | 95.0% | 97.1% | 97.1% | 96.4% | 95.3% | 101.4% | |
| Retain 160 Tokens (↓ **94.4%**) | | | | | | | |
| SparseVLM (ICML'25) | 51.2 | 63.1 | 1542 | 67.5 | 46.4 | 32.8 | 86.9% |
| | 79.8% | 92.9% | 83.7% | 96.2% | 75.7% | 93.4% | |
| VisionZip (CVPR'25) | 55.5 | 60.1 | 1630 | 68.3 | 56.2 | 36.1 | 92.5% |
| | 86.4% | 88.5% | 88.5% | 97.3% | 91.7% | 102.8% | |
| Ours | 60.0 | 64.3 | 1700 | 67.4 | 56.8 | 35.6 | 95.1% (↓ **4.9%**) |
| | 93.5% | 94.7% | 92.3% | 96.0% | 92.7% | 101.4% | |

## 4.2 Main Results

**Results on LLaVA 1.5.** LLaVA 1.5 is one of the most representative MLLMs. We therefore apply the proposed pruning method to LLaVA 1.5 and evaluate its performance on a variety of image understanding tasks, following prior works [41, 49, 43]. Due to the diverse evaluation metrics used across different benchmarks, which result in inconsistent numerical scales, we report performance as a percentage of the original model's accuracy. We show the results of LLaVA 1.5 7B in Table 1. In particular, we follow previous work [49, 43] and evaluate the performance under three visual token pruning budgets (*i.e.*, 192, 128, and 64) to evaluate the effectiveness of the proposed method. The vanilla model (*i.e.*, LLaVA 1.5 7B with full visual tokens) serves as the upper bound (100%), representing the performance ceiling of any visual token pruning approach. Our method consistently outperforms existing approaches across all token configurations, particularly under aggressive compression settings. As shown in Table1, when retaining only 192 tokens (a 66.7% reduction from the baseline), our method achieves an average accuracy of 99.5% relative to the upper bound. This surpasses state-of-the-art baselines including FastV [7] (+6.0%), SparseVLM[49] (+3.0%), and VisionZip [43] (+1.5%). Under extreme compression (*e.g.*, 64 tokens, 88.9% reduction), our method maintains 96.0% of the original performance, significantly outperforming baselines such as VisionZip [43] (93.5%) and SparseVLM [49] (85.1%).

Surprisingly, our method preserves or even surpasses the upper bound in performance on several benchmarks. For instance, we observe relative accuracies of 100.2% and 104.5% on POPE [22] and MMVet [45], respectively, when using 192 tokens. These results suggest that visual tokens in MLLMs contain redundancy, and our method not only reduces this redundancy but also improves performance by eliminating interference from redundant information. We further evaluate our method on the larger LLaVA 1.5 13B model to validate its generalization capability in Appendix C.1.

**Results on LLaVA-Next.** Compared to LLaVA 1.5, LLaVA-Next is a more advanced MLLM that supports high-resolution image processing, thereby significantly improving vision-language understanding. LLaVA-Next partitions an input image into multiple regions based on its original size. Usually, the image is divided into 4 sub-images. Both the original and partitioned images are then encoded into visual tokens, resulting in a total of 2,880 tokens (576×5). While effective in capturing fine-grained visual details, this strategy substantially increases the number of visual

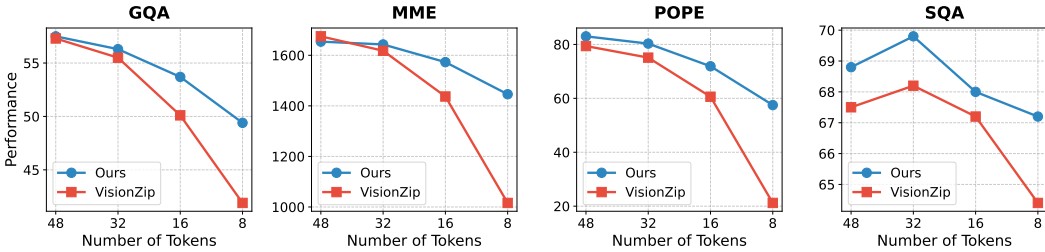

Figure 4: The performance comparison under extreme token number.

tokens and reduces inference efficiency. Therefore, our objective is to minimize the number of visual tokens while maintaining performance as much as possible. To evaluate the proposed method on LLaVA-Next, we follow previous works [49, 43] and adopt three visual token budget settings (*i.e.*, 640, 320, and 160). The results are presented in Table 2. As shown, our method consistently outperforms state-of-the-art approaches under all configurations. Specifically, when retaining only 640 tokens, our approach achieves an average accuracy of 98.9% relative to the upper bound. Under extreme compression (*e.g.*, 160 tokens, 94.4% reduction), our method maintains 95.1% performance, significantly surpassing baselines such as SparseVLM (86.9%) and VisionZip (92.5%). These results further validate the effectiveness of the proposed method across different MLLM architectures. We also evaluate our method on the LLaVA-Next 13B model in Appendix C.1.

**Results on Video benchmarks.** We further evaluate the effectiveness of the proposed method, and we implement our SCOPE based on Video-LLaVA following VisionZIP [43]. The results are reported in Table 3. As shown, our method achieves the best performance among all compared methods. Surprisingly, even with aggressive pruning, our method almost fully preserves the original performance. This demonstrates the strong effectiveness of our method on video-language tasks. These findings also

Table 3: Performance comparison on Video-LLaVA. The original Video-LLaVA's video token number is 2048, while our method only retains the 136 tokens.

| Method | TGIF | MSVD | MSRVTT | ActivityNet | Avg |
|---|---|---|---|---|---|
| Video-LLaVA | 47.1 | 69.8 | 56.7 | 43.1 | 100.0% |
| FastV | 23.1 | 38.0 | 19.3 | 30.6 | 52.1% |
| | 49.0% | 54.4% | 34.0% | 71.0% | |
| SparseVLM | 44.7 | 68.2 | 31.0 | 42.6 | 86.5% |
| | 94.9% | 97.7% | 54.7% | 98.8% | |
| VisionZip | 42.4 | 63.5 | 52.1 | 43.0 | 93.2% |
| | 90.0% | 91.0% | 91.9% | 99.8% | |
| Ours | 47.1 | 69.2 | 55.9 | 44.9 | **100.5%** |
| | 100.0% | 99.1% | 98.6% | 104.2% | |

suggest that video benchmarks contain substantial redundancy, and token pruning has great potential for accelerating video LLMs without sacrificing performance.

## 4.3 Analysis

**Results under Extreme Token Reduction.** Our method demonstrates superior performance stability as the number of visual tokens is progressively reduced. As shown in Fig. 4, even when the token count is reduced to as few as 8, our approach consistently outperforms VisionZip [43] by increasingly larger margins. This highlights the strong capability of our framework to retain critical visual information under extreme compression. In contrast, VisionZip exhibits a sharp performance drop in low-token regimes, further validating the effectiveness of our token selection strategy and underscoring the potential of our method for aggressive visual token pruning.

**Ablation Studies.** As shown in Table 4, our method, which jointly considers token saliency and coverage, consistently outperforms its ablated variants (saliency-only and coverage-only) across all benchmarks. Both ablated models still perform better than the

Table 4: Ablation studies of the proposed method.

| | GQA | MMB | MME | POPE | TextVQA |
|---|---|---|---|---|---|
| Random | 55.5 | 54.0 | 1556 | 75.2 | 48.4 |
| Saliency-only | 55.0 | 60.8 | 1665 | 76.8 | 55.4 |
| Coverage-only | 58.1 | 60.8 | 1687 | 82.1 | 56.3 |
| Ours | **58.3** | **61.7** | **1698** | **83.9** | **56.6** |

random baseline, indicating the individual effectiveness of each component. For instance, the coverage-only variant achieves moderate performance. However, our full method further improves these results, demonstrating that combining saliency and coverage provides complementary benefits. Explicit modeling of both saliency and coverage leads to superior performance compared to using either criterion alone or selecting tokens randomly.

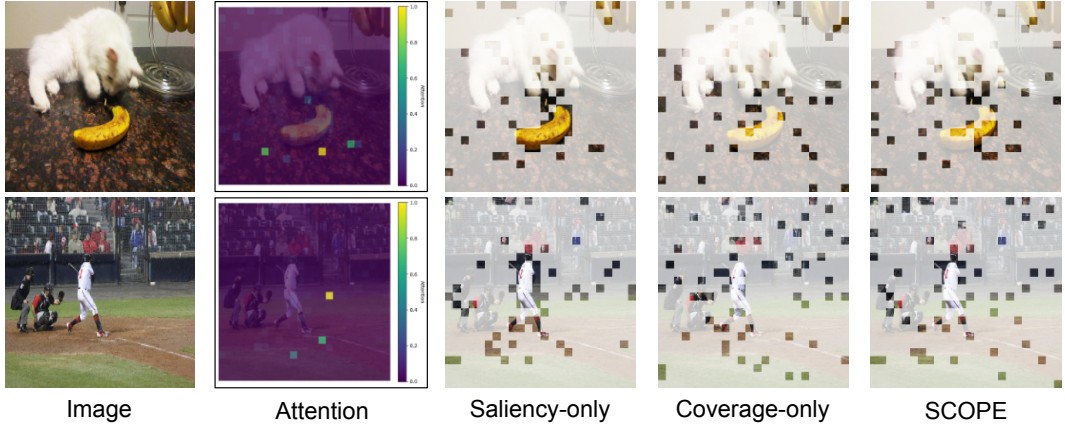

| Image | Attention | Saliency-only | Coverage-only | SCOPE |

Figure 5: Visualization of token pruning among different pruning strategies.

**Efficiency Analysis.** Table 5 compares the efficiency of our method with that of a baseline pruning approach (PDrop) on LLaVA-NeXT 7B. Despite reducing the number of visual tokens from 2,880 to 160, a compression ratio exceeding $18\times$, our method maintains strong performance on the POPE metric (81.3% vs. 86.4%), demonstrating

Table 5: Efficiency analysis of our method on LLaVA-NeXT 7B. The experiments are conducted on a system equipped with $4\times$A100. $\Delta$ denotes the reduction ratio.

|         | Token Number | POPE | Latency (s) | $\Delta$ |
|---------|--------------|------|-------------|----------|
| Vanilla | 2880         | 86.4 | 601.9       | -        |
| PDrop   | 160          | 53.2 | 184.0       | $3.3\times$ |
| Ours    | 160          | 81.3 | 188.8       | $3.2\times$ |

that our token selection strategy effectively preserves semantic completeness. In contrast, PDrop [41] exhibits a substantial performance drop (53.2%), likely due to its reliance on saliency-based pruning, which may overlook less attended yet semantically important regions. Although our method incurs slightly higher latency than PDrop, it still achieves a $3.2\times$ speedup over the full-token baseline. This indicates that our saliency-coverage oriented pruning strategy is not only effective in preserving performance but also computationally efficient in practice.

**Token Pruning Visualization.** In Fig. 5, we provide a visualization of token pruning to illustrate the difference of selected tokens among different strategies. Saliency-only mainly concentrates on the most salient patch such as the cat and banana in 1st row, demonstrating object-level focus by pruning the background. Coverage-only selects the tokens that are spread across the image, preserving global context but potentially missing important object details. Our SCOPE maintains the high token density on salient patches (*e.g.*, cat and banana in 1st row), while a sparse set of tokens is strategically kept for the background. This captures critical object features without discarding essential scene context.

## 5 Conclusion

While existing approaches predominantly rely on attention-based saliency to prune redundant tokens, they often neglect semantic coverage, leading to incomplete visual representations. To overcome this limitation, we propose SCOPE, a novel visual token pruning framework that jointly models both token saliency and coverage. Our method introduces a set-coverage score based on pairwise token similarities and calculates a token-coverage gain for each candidate token. By incorporating saliency scores into this gain, we derive the SCOPE score, which guides an iterative token selection process. Empirical evaluations on LLaVA 1.5 and LLaVA-Next across multiple vision-language benchmarks show that SCOPE consistently outperforms state-of-the-art pruning approaches, achieving strong performance even under aggressive token reduction. We believe that our approach offers a principled and effective framework for evaluating the value of visual tokens in MLLMs.

## Acknowledgement

This work was supported by National Natural Science Foundation of China (No.62476051, No.62176047, No.82121003) and Sichuan Natural Science Foundation (No.2024NSFTD0041). This research is supported in part by the National Research Foundation, Singapore under its National Large Language Models Funding Initiative (AISG Award No: AISG-NMLP-2024-003), in part by A*STAR Career Development Fund (CDF) under C243512011. Any opinions, findings and conclusions or recommendations expressed in this material are those of the author(s) and do not reflect the views of National Research Foundation, Singapore.

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

# A   Overview

This appendix provides detailed information on the experimental benchmarks, additional qualitative results, and visualizations that support the main claims of the paper. In SectionB, we present comprehensive descriptions of the benchmark datasets. Section C includes supplementary experiments, such as results on larger models (LLaVA 1.5 13B and LLaVA-Next 13B), as well as a hyperparameter analysis. In Section D, we provide additional visualization studies to further illustrate the behavior of our method. Finally, we discuss the broader impact and limitations of our work.

# B   Benchmarks

We conduct the experiments on several widely used visual understanding benchmarks. In the following, we will give a detailed description of these benchmarks.

**GQA.** [13]. The GQA benchmark consists of three components: scene graphs, questions, and images. The image component includes raw images, their spatial features, and the features of all objects within the images. The questions in GQA are crafted to evaluate visual scene understanding and reasoning about various aspects of an image. Our method is evaluated on the subset of "testdev_balanced_instructions", which includes 12,578 samples.

**MMBench.** [27]. MMBench is a comprehensive benchmark designed to evaluate the multi-modal capabilities of large language models, covering a wide range of tasks including visual question answering, image captioning, cross-modal retrieval, and creative generation. It provides a fine-grained assessment from perception to cognition, containing approximately 3,000 multiple-choice questions aggregated from diverse sources. The benchmark aims to measure whether a model is a true "all-around player" in multi-modal understanding and reasoning.

**MME.** [12]. The MME benchmark is a comprehensive evaluation suite carefully crafted to assess multiple facets of model performance. It comprises 14 distinct subtasks targeting both perceptual and cognitive capabilities of models. By employing manually curated instruction-answer pairs and succinct instruction formats, MME effectively reduces the risks of data leakage and ensures a fairer assessment of model abilities. We evaluate the performance on the dev split including 4,377 samples. The evaluation metric is the accuracy of the model's answer.

**POPE.** [22] The POPE benchmark focuses on assessing object hallucination in models by presenting them with a set of targeted yes/no questions about object existence within images. This approach reframes the evaluation of hallucination, emphasizing the model's ability to correctly identify whether certain objects are present. To quantitatively analyze performance across three distinct sampling methods, the benchmark utilizes metrics such as accuracy, recall, precision, and F1 score, offering a robust measure of the model's susceptibility to hallucination. We evaluate the model's performance on the test split, including 9,000 samples. The evaluation metric is the F1 score.

**ScienceQA (SQA).** [29] Encompassing a wide array of fields such as natural sciences, linguistics, and social sciences, SQA structures its questions through a hierarchical framework consisting of 26 topics, 127 categories, and 379 distinct skills. This benchmark is designed to rigorously test a model's proficiency in multimodal comprehension, complex reasoning across multiple steps, and interpretability. By organizing questions first by subject area, then by specific category, and finally by the required skill, SQA ensures a thorough and nuanced assessment of scientific understanding across diverse domains. This layered organization enables a detailed evaluation of a model's ability to handle a broad spectrum of scientific queries. The evaluation metric is the accuracy.

**TextVQA.** [36] TextVQA is designed to assess a model's capability to interpret and reason over textual content embedded in images. This benchmark challenges models with visual question answering tasks that require both comprehension of image context and accurate reading of the text present within the images. To perform well, models must effectively integrate visual and textual cues, demonstrating robust understanding and reasoning skills related to text in complex visual environments. We evaluate the model's performance on the test split, including 5,000 samples. The evaluation metric is exact match (EM).

**SEEDBench** [18] SEEDBench features a collection of 19,000 multiple-choice questions curated by human annotators. Covering 12 different evaluation dimensions, this benchmark examines models'

Table 6: **Performance comparison under different vision token configurations.** The evaluated model is LLaVA 1.5 13B, where the default number of visual tokens is 576. The first row for each method reports the raw accuracy across benchmarks, and the second row indicates the performance relative to the upper bound.

| Method | GQA | MMB | MME | POPE | SQA | TextVQA | SEED-I | MMVet | Avg. |
|---|---|---|---|---|---|---|---|---|---|
| Upper Bound, 576 Tokens (100%) | | | | | | | | | |
| Vanilla (CVPR'24) | 63.2 | 67.7 | 1818 | 85.9 | 72.8 | 61.3 | 66.9 | 35.3 | 100% |
| | 100% | 100% | 100% | 100% | 100% | 100% | 100% | 100% | |
| Retain 192 Tokens (↓ 66.7%) | | | | | | | | | |
| VisionZip (CVPR'25) | 59.1 | 66.9 | 1754 | 85.1 | 73.5 | 59.5 | 65.2 | 37.5 | 98.7% |
| | 93.5% | 98.8% | 96.5% | 99.1% | 101.0% | 97.1% | 97.5% | 106.20% | |
| Ours | 59.7 | 67.6 | 1775 | 86.7 | 73.8 | 60 | 65.5 | 39.4 | 100.2% |
| | 94.5% | 99.9% | 97.6% | 100.9% | 101.4% | 97.9% | 97.9% | 111.6% | |
| Retain 128 Tokens (↓ 77.8%) | | | | | | | | | |
| VisionZip (CVPR'25) | 57.9 | 66.7 | 1743 | 85.2 | 74 | 58.7 | 63.8 | 37.5 | 97.0% |
| | 91.6% | 98.5% | 95.9% | 99.2% | 101.6% | 95.8% | 95.4% | 106.2% | |
| Ours | 59.3 | 67.2 | 1735 | 85.9 | 73.9 | 58.7 | 64.8 | 37.7 | 98.7% |
| | 93.8% | 99.3% | 95.4% | 100.0% | 101.5% | 95.8% | 96.9% | 106.8% | |
| Retain 64 Tokens (↓ 88.9%) | | | | | | | | | |
| VisionZip (CVPR'25) | 56.2 | 64.9 | 1676 | 76.0 | 74.4 | 57.4 | 60.4 | 33.9 | 93.7% |
| | 88.9% | 95.9% | 92.2% | 88.5% | 102.2% | 93.3% | 90.3% | 96.0% | |
| Ours | 58.7 | 65.5 | 1762 | 83.0 | 73.2 | 58.3 | 63.6 | 35.7 | 96.9% |
| | 92.9% | 96.8% | 96.9% | 96.6% | 100.5% | 95.1% | 95.1% | 101.1% | |

capabilities in identifying patterns within both images and videos, taking into account spatial as well as temporal characteristics. The evaluation metric is the accuracy.

**MMVet** [45] The MMVet benchmark is constructed with the understanding that tackling complex tasks typically requires a generalist model to effectively combine multiple fundamental vision-language skills. MMVet identifies six essential vision-language capabilities and systematically evaluates sixteen specific combinations arising from these core abilities, thereby assessing the model's proficiency in integrating diverse vision-language functions. We evaluate the model's performance on the test split, including 218 samples. The score is evaluated by the GPT model.

## C  Additional Experiments

In the main paper, we present experiments on LLaVA 1.5 7B and LLaVA-Next 7B. To further demonstrate the generalizability of our method across model scales, we provide additional results on LLaVA 1.5 13B and LLaVA-Next 13B. We also provide the results on more MLLMs such as Qwen2-VL and more OCR-related benchmarks.

### C.1  Results on LLaVA 1.5 13B

As shown in Table 6, our method consistently outperforms VisionZip [43] across all token budgets. With 192 tokens, our approach achieves 100.2% of the upper bound's average performance, slightly higher than VisionZip [43] (98.7%). The advantage becomes more evident as the token count decreases: at 64 tokens, our method retains 96.9% performance, compared to VisionZip's 93.7%. Notably, on benchmarks like MMVet [45] and POPE [22], our method even surpasses the original model's performance. These results demonstrate that our joint saliency-coverage strategy better preserves essential information under aggressive token pruning.

### C.2  Results on LLaVA-Next 13B

We present the results on LLaVA-Next 13B in Table 7. We can observe that our method consistently outperforms VisionZip [43] under all token budgets. For example, with 640 tokens, our approach

Table 7: **Performance comparison under different vision token configurations.** The evaluated model is LLaVA-Next 13B. The vanilla number of vision tokens is $2,880$. The first line of each method is the raw accuracy on the benchmarks, and the second line is the proportion relative to the upper bound.

| Method | GQA | MMB | MME | POPE | SQA | TextVQA | MMMU | SEED-I | Avg. |
|---|---|---|---|---|---|---|---|---|---|
| Upper Bound, 2880 Tokens (100%) | | | | | | | | | |
| Vanilla 13B (CVPR'24) | 65.4 | 70.0 | 1901 | 86.2 | 73.5 | 64.3 | 36.2 | 71.9 | 100% |
| | 100% | 100% | 100% | 100% | 100% | 100% | 100% | 100% | |
| Vanilla 7B (CVPR'24) | 64.2 | 67.9 | 1842 | 86.4 | 70.2 | 61.3 | 35.1 | 70.2 | 97.2% |
| | 98.2% | 97.0% | 96.9% | 100.2% | 95.5% | 95.3% | 97.0% | 97.6% | |
| Retain 640 Tokens (↓ **77.8%**) | | | | | | | | | |
| **VisionZip** (CVPR'25) | 63.0 | 68.6 | 1871 | 85.7 | 71.2 | 62.2 | 36.4 | 68.8 | 97.8% |
| | 96.3% | 98.0% | 98.4% | 99.4% | 96.9% | 96.7% | 100.6% | 95.7% | |
| Ours | 63.6 | 69.3 | 1897 | 86.4 | 72.5 | 62.4 | 36.6 | 69.9 | 98.8% |
| | 97.2% | 99.0% | 99.8% | 100.2% | 98.6% | 97.0% | 101.1% | 97.2% | |
| Retain 320 Tokens (↓ **88.9%**) | | | | | | | | | |
| VisionZip (CVPR'25) | 60.7 | 67.2 | 1805 | 82.0 | 70.3 | 60.9 | 35.6 | 65.2 | 94.8% |
| | 92.8% | 96.0% | 95.0% | 95.1% | 95.6% | 94.7% | 98.3% | 90.7% | |
| Ours | 63.0 | 67.7 | 1830 | 85.1 | 71.7 | 60.8 | 36.3 | 67.9 | 96.9% |
| | 96.3% | 96.7% | 96.3% | 98.7% | 97.6% | 94.6% | 100.3% | 94.4% | |
| Retain 160 Tokens (↓ **94.4%**) | | | | | | | | | |
| VisionZip (CVPR'25) | 57.8 | 64.9 | 1739 | 76.6 | 69.3 | 58.4 | 37.0 | 61.1 | 91.7% |
| | 88.4% | 92.7% | 91.5% | 88.9% | 94.3% | 90.8% | 102.2% | 85.0% | |
| Ours | 61.4 | 66.9 | 1777 | 82.8 | 72.0 | 59.3 | 36.2 | 66.1 | 95.1% |
| | 93.9% | 95.6% | 93.5% | 96.1% | 98.0% | 92.2% | 100.0% | 91.9% | |

achieves 98.8% of the upper bound's average performance, compared to VisionZip's 97.8%. As the token count decreases to 160, our method still retains 95.1% performance, while VisionZip drops to 91.7%. These results further confirm the superior robustness of our method under aggressive token pruning.

## C.3 Results on Qwen2-VL

To further evaluate the generalization of the proposed SCOPE, we have also evaluated our method on the Qwen2-VL [39] model. The results are summarized in Table 8. As shown, our method achieves 94.6% of the full-model performance when retaining only 25% of the tokens. Furthermore, our method significantly outperforms prior approaches such as DivPrune [2], with a 3.7% improvement in average score under the 10.0% token ratio.

Table 8: **Results on Qwen2-VL.** The token ratio means the ratio of retained tokens.

| Method | Token Ratio | GQA | MMB | MME | POPE | Avg. |
|---|---|---|---|---|---|---|
| Qwen2-VL 7B | 100% | 61.9 | 77.4 | 2286 | 88.4 | 100% |
| DivPrune | 25% | 59.4 | 72 | 2043 | 85.9 | 93.90% |
| Ours | 25% | 59.8 | 72.5 | 2065 | 86.5 | 94.6%(+0.7%) |
| DivPrune | 10% | 54.3 | 63.7 | 1874 | 80.8 | 85.90% |
| Ours | 10% | 56.6 | 66.8 | 1953 | 84.3 | 89.6%(+3.7%) |

## C.4 Results on more OCR Benchmarks.

In the main paper, we have already evaluated our method on several OCR-related benchmarks, such as MME [12] and MMVet [45]. To further demonstrate the effectiveness of SCOPE, we conducted additional experiments on more OCR-specific benchmarks including DocVQA [32], ChartQA [31]

and OCRBench [28]. The results are shown in Table 9. As illustrated, our method consistently preserves performance and outperforms VisionZip across different token counts. This further supports the robustness of our approach for OCR tasks.

Table 9: **Results on more OCR Benchmarks.** The model is LLaVA 1.5 7B.

| Method | #Token | DocVQA | ChartQA | OCRBench | Avg. |
|---|---|---|---|---|---|
| Vanilla | 576 | 28.0 | 18.2 | 31.3 | 100% |
| VisionZip | 192 | 26.0 | 17.3 | 31.1 | 95.8% |
| Ours | 192 | 26.5 | 17.4 | 31.2 | 96.6% (+0.8%) |
| VisionZip | 128 | 25.1 | 17.1 | 30.0 | 93.1% |
| Ours | 128 | 25.9 | 17.3 | 30.7 | 95.2% (+2.1%) |
| VisionZip | 64 | 21.1 | 16.0 | 28.2 | 84.4% |
| Ours | 64 | 23.2 | 16.7 | 29.5 | 89.6%(+5.2%) |

## C.5 Hyper-parameter Analysis

The hyperparameter $\alpha$ controls the scaling of the attention scores, thereby influencing token selection in our method. As illustrated in Fig. 6, the optimal performance is typically achieved when $\alpha = 1.0$, suggesting that this setting effectively balances saliency and coverage across most benchmarks.

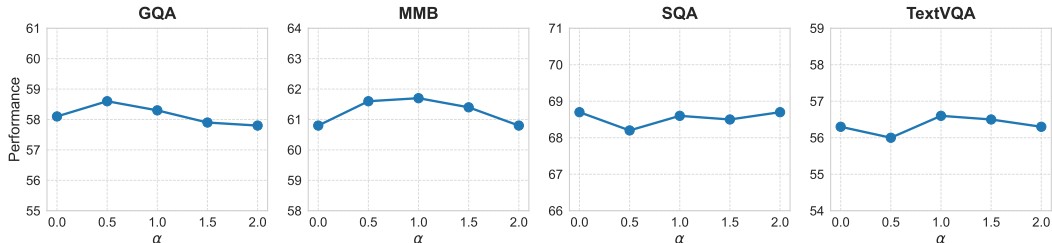

Figure 6: The hyperparameter $\alpha$ analysis on LLaVA 1.5 7B with 64 visual tokens.

## D    Visualization Results

We present additional results on selected token visualization in Fig. 7. The saliency-based method selects tokens solely based on attention scores, which may overlook semantically important tokens that contribute to the overall completeness of the visual representation. In contrast, our saliency-coverage oriented approach jointly considers both visual saliency and semantic coverage. As a result, the selected tokens span a broader region in the embedding space.

In Fig. 8, we further visualize the attention distribution of selected tokens. Our method preserves the majority of high-attention tokens, demonstrating its ability to retain both informative and representative visual content.

## E    Broader Impact

Our proposed method aims to improve both the efficiency and effectiveness of multimodal large language models (MLLMs) by reducing the number of visual tokens while preserving semantic completeness. This advancement has the potential to significantly reduce the computational cost and memory footprint of MLLMs, thereby enhancing their feasibility for deployment in resource-constrained environments such as edge devices, mobile platforms, and real-time applications. By enabling more efficient inference, our approach can facilitate the broader adoption of vision-language models across various domains, including education, healthcare, and assistive technologies.

However, as with any technology that enhances the scalability and accessibility of AI systems, there are potential societal risks. For example, more efficient MLLMs could be misused to generate or disseminate misinformation, enable invasive surveillance, or support other malicious activities,

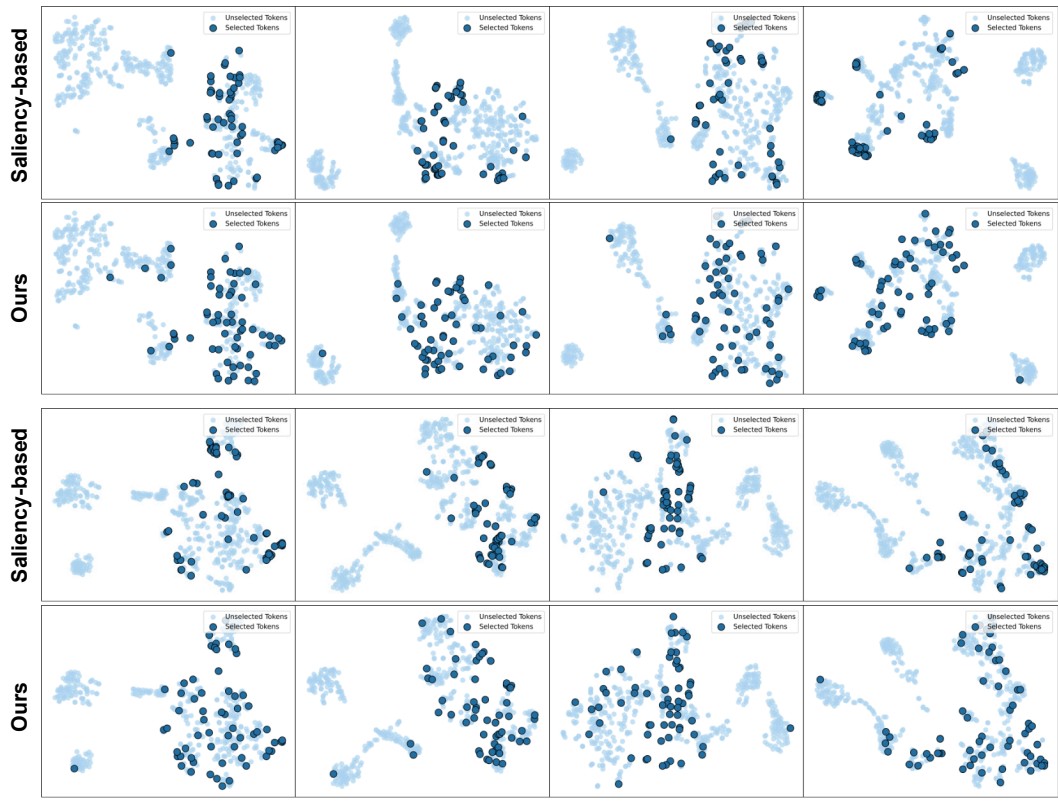

Figure 7: The selected token comparison between the saliency-based method and our saliency-coverage oriented method. The total visual token number is 576, and the selected token number is 64.

particularly when deployed at scale. It is therefore essential to consider these ethical implications and implement appropriate safeguards when deploying such models in practice.

## F Limitations

While SCOPE demonstrates strong performance and efficiency gains across multiple benchmarks and model architectures, several limitations remain. (1) Despite our efforts to balance saliency and coverage, aggressive token pruning may still result in the loss of fine-grained or rare semantic information, potentially affecting tasks that require detailed visual understanding. (2) Our experiments are primarily based on widely used vision-language benchmarks and two representative MLLMs, LLaVA 1.5 and LLaVA-Next. Therefore, the generalizability of SCOPE to other tasks or model architectures has yet to be fully validated.

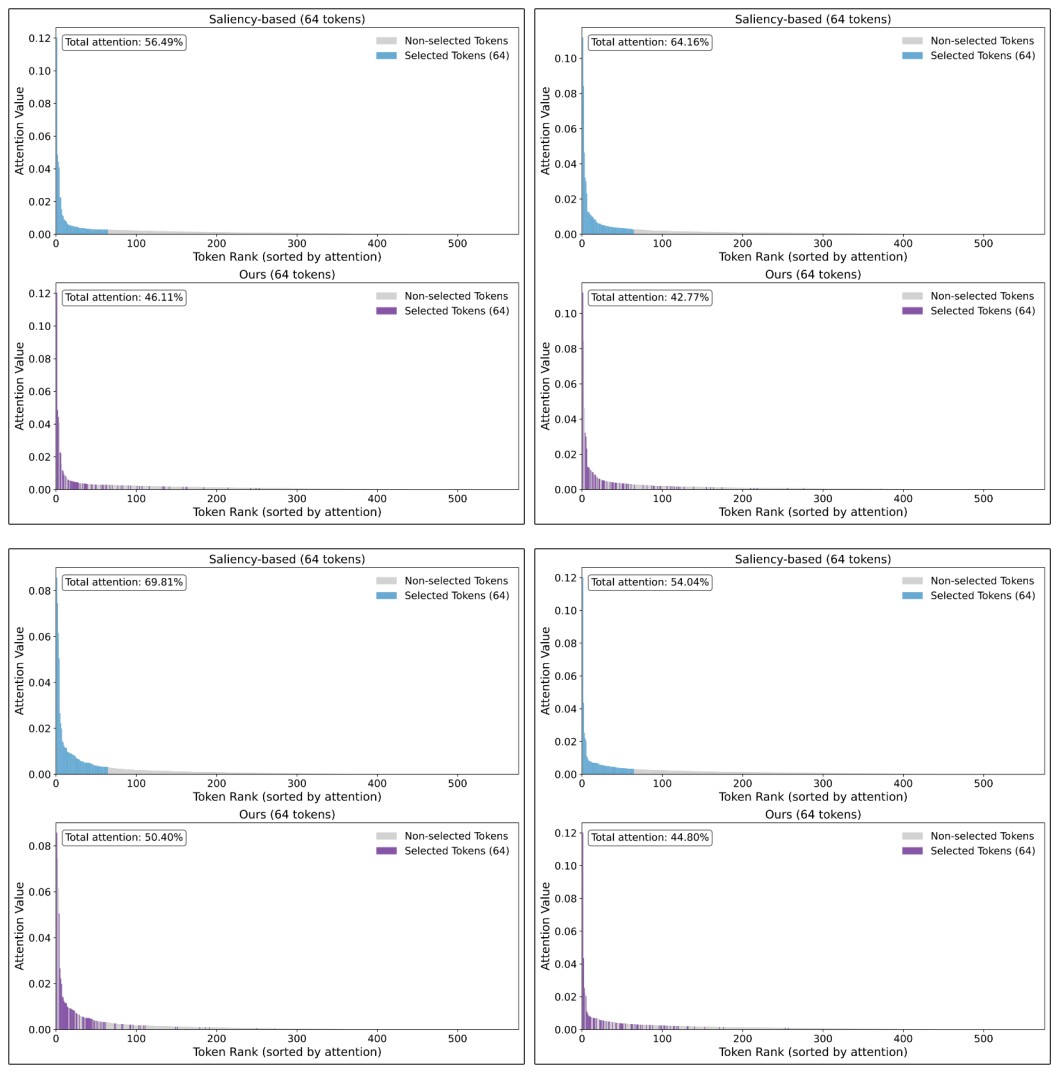

Figure 8: Attention distribution visualization for selected token. The total visual token number is 576, and the selected token number is 64. Our method retained most of the high attention tokens and some low attention tokens to maximize the coverage.

