# OpenReview forum: "SCOPE: Saliency-Coverage Oriented Token Pruning for Efficient Multimodel LLMs"
_NeurIPS.cc/2025/Conference — NeurIPS 2025 poster_

### Official Review · Reviewer_6ZPM · 2025-06-27

**Clarity:** 2
**Significance:** 2
**Originality:** 2
**Rating:** 4
**Confidence:** 5

**Summary:**

This paper proposes a saliency-coverage oriented token pruning method for efficient VLMs, which jointly model both the saliency and coverage to measure the redundancy of visual tokens as SCOPE scores. Experiments on LLaVA-1.5 and LLaVA-Next models demonstrate the validity of the proposed method.

**Questions:**

1. Why compute the SCOPE score with the product of visual attention score and coverage gain? Have you ever tested adding these two values? Is there an ablation study?
2. I would like to see more results with advanced architectures (qwen2-vl) or video benchmarks, OCR tasks to see the generalizability of SCOPE for different models and tasks.

**Ethical Concerns:**

["NO or VERY MINOR ethics concerns only"]

**Final Justification:**

The rebuttal address my concerns and I raise the score to Borderline Accept.

**Quality:**

2

**Strengths And Weaknesses:**

Strengths:
1. The paper is well-written and easy to understand.
2. The proposed method is simple and intuitive for measuring the redundancy of visual tokens for efficient VLMs.
3. Experimental results demonstrate the effectiveness of the proposed method to some extent.

Weaknesses:
1. The novelty of the proposed method is limited. The "saliency" is just the attention score which has been widely adopted in previous methods. Besides, I think the proposed "coverage" measure is quite similar to the diversity, which is also calculated with the cosine similarity, such as Divprune [1].

2. The proposed method should be evaluated on SOTA models to demonstrate the effectiveness, such as qwen2-vl. Besides, could the proposed method be combined with LLM-based pruning methods, such as FastV ? The proposed method should also be tested on video benchmarks to prove the validity.

3. I am interested in some OCR benchmarks, which demand more advanced token pruning methods for maintaining the performance.

[1] DivPrune: Diversity-based Visual Token Pruning for Large Multimodal Models. CVPR2025.

---

> ### Author Rebuttal · Authors · 2025-07-31
>
> We sincerely thank the reviewer for their positive feedback on our paper, particularly for recognizing its clarity, the simplicity and intuitiveness of our proposed method. We greatly appreciate these encouraging comments. In the following, we will address each of the reviewer’s concerns and suggestions point-by-point to further clarify and strengthen our work.
>
> > **Q1**: The novelty of the proposed method is limited. The "saliency" is just the attention score which has been widely adopted in previous methods. Besides, I think the proposed "coverage" measure is quite similar to the diversity, which is also calculated with the cosine similarity, such as Divprune [1].
>
> **A1**: Thank you for your comment. We respectfully clarify that our method is distinctive from existing works, with the following contributions:
>
> 1. **Coverage is different from diversity**: While both coverage and diversity involve cosine similarity, they serve fundamentally different purposes. For example, DivPrune [1] focuses on selecting tokens that are mutually dissimilar, aiming to maximize pairwise diversity among selected tokens. In contrast, our coverage-based selection encourages selecting tokens that best represent the entire token set by maximizing set coverage gain. As illustrated in Figure D1, DivPrune would select token C to maximize diversity from already selected tokens, while our coverage strategy selects token B because it better covers the unselected tokens A and C (i.e., provides broader representativeness). This distinction is also supported by empirical results, for example, our coverage-only baseline achieves 56.3% on TextVQA and 60.8% on MMB, outperforming DivPrune’s 54.9 and 59.3%, respectively.
>
> **Figure D1: Illustration of token selection. ● and ○ selected and unselected tokens.**
> ```
>   ●   ●
> ●       ●    ○ A    ○ B    ○ C
>     ●               ↑      ↑
>                 Coverage Diversity
> ```
>
> 2. **We are the first to jointly consider saliency and coverage**: While saliency (e.g., attention scores) has been used in previous token pruning methods, these approaches often lead to semantic incompleteness by focusing only on highly attended regions. In contrast, our method, SCOPE, formulates token selection as a joint optimization over both saliency and coverage. This ensures that selected tokens are not only important (salient) but also representative of the full image (coverage), leading to improved semantic completeness and task performance.
>
> We hope this clarifies the novelty of our approach. We will further improve the explanation and emphasize this distinction in the final version.
>
>
> > **Q2**: The proposed method should be evaluated on SOTA models to demonstrate the effectiveness, such as qwen2-vl.
>
> **A2**: Thank you for the suggestion. Following your advice, we have evaluated our method on the Qwen2-VL model. The results are summarized in Table D1. As shown, our method achieves 94.6% of the full-model performance when retaining only 25% of the tokens. Furthermore, our method significantly outperforms prior approaches such as DivPrune, with a 3.7% improvement in average score under the 10% token ratio.
>
> We will include these results in the final version to demonstrate the robustness and generality of our approach on strong vision-language models.
>
> **Table D1: Results on Qwen2-VL. The token ratio means the ratio of retained tokens.**
> |Method|Token Ratio|GQA|MMB|MME|POPE|Avg.|
> |:-:|:-:|:-:|:-:|:-:|:-:|:-:|
> |Qwen2-VL 7B | 100% | 61.9|	77.4|	2286|	88.4|100%|
> |DivPrune | 25% | 59.4	|72	|2043	|85.9 | 93.9%|
> |**Ours** | 25% | 59.8 |72.5| 2065 | 86.5|94.6% (**+0.7%**)|
> |DivPrune | 10% | 54.3|	63.7|1874|80.8 |85.9%|
> |**Ours** | 10% | 56.6 | 66.8 | 1953 | 84.3 | 89.6% (**+3.7%**)|
>
>
> > **Q3**: Besides, could the proposed method be combined with LLM-based pruning methods, such as FastV?
>
> **A3**: Thank you for the valuable suggestion. We have explored the possibility of combining our SCOPE method with LLM-based pruning approaches such as FastV. In particular, we first apply SCOPE to prune visual tokens before feeding them into the LLM, and then apply FastV for further pruning within the language model. The results are provided in Table D2. These results indicate that SCOPE can be effectively combined with FastV to further reduce the number of visual tokens while maintaining reasonable performance. However, there remains significant room for improvement, and we consider this a promising direction for future exploration.
>
> **Table D2: The results of SCOPE combined with FastV. We first retain 192 tokens and then prune R ratio tokens at K-th layers of LMM using FastV.**
> |Method | GQA | MMB | MME | TextQA| Avg.|
> |:-:|:-:|:-:|:-:|:-:|:-:|
> |Vanilla | 61.9	|64.7	|1862 |58.2| 100%|
> |SCOPE (192) | 60.1|	63.6 |1804 |57.7| 97.8% |
> |SCOPE (192)+ FastV (K=3, R=50%) | 57.1|	61.8| 1712| 56.2| 94.1% |
>
> > **Q4**: The proposed method should also be tested on video benchmarks to prove the validity.
>
> **A4**: Thank you for the suggestion. Following VisionZip, we conducted experiments on the Video-LLaVA. The results are reported in Table D3. As shown, our method achieves the best performance among all compared methods. Surprisingly, even with aggressive pruning, our method almost fully preserves the original performance. This demonstrates the strong effectiveness of our method on video-language tasks. These findings also suggest that video benchmarks contain substantial redundancy, and token pruning has great potential for accelerating video LLMs without sacrificing performance. We will include these results in the final version.
>
> **Table D3: Results on video benchmarks.**
> | Method       |#Token| TGIF | MSVD | MSRVTT | ActivityNet | Avg.     |
> |--------------|:-:|:-:|:-:|:-:|:-:|:-:|
> | Video-LLaVA  |256\*8| 47.1 | 69.8 | 56.7   | 43.1        | 100.0%  |
> | FastV        |17\*8| 23.1 | 38.0 | 19.3   | 30.6        | 52.1%   |
> | SparseVLM    |17\*8| 44.7 | 68.2 | 31.0   | 42.6        | 86.5%   |
> | VisionZip    |17\*8| 42.4 | 63.5 | 52.1   | 43.0        | 93.2%  |
> | **Ours**         |17\*8| 47.1 | 69.2 | 55.9   | 44.9        | **100.5%** |
>
>
> > **Q5**: I am interested in some OCR benchmarks, which demand more advanced token pruning methods for maintaining the performance.
>
> **A5**: Thank you for your suggestion. In the main paper, we have already evaluated our method on several OCR-related benchmarks, such as MME and MMVet. To further demonstrate the effectiveness of SCOPE, we conducted additional experiments on more OCR-specific benchmarks. The results are shown in Table D4. As illustrated, our method consistently preserves performance and outperforms VisionZip across different token counts. This further supports the robustness of our approach for OCR tasks. We will include these results in the final version.
>
> **Table D4: More results on OCR benchmarks.**
> |Method|#Token|DocVQA|ChartQA|OCRBench|Avg.|
> |:-:|:-:|:-:|:-:|:-:|:-|
> |Vanilla | 576 | 28.0 | 18.2 | 31.3 |100%|
> |VisionZip| 192 | 26.0 | 17.3 | 31.1 |95.8%|
> |**Ours** | 192| 26.5 | 17.4 | 31.2 |96.6% (**+0.8%**)|
> |VisionZip|128| 25.1 | 17.1 | 30.0 |93.1%|
> |**Ours** | 128| 25.9 | 17.3 | 30.7 |95.2% (**+2.1%**)|
> |VisionZip|64|21.1| 16.0 | 28.2 |84.4%|
> |**Ours** | 64 | 23.2 | 16.7 |29.5 |89.6% (**+5.2%**)|
>
> > **Q6**: Why compute the SCOPE score with the product of visual attention score and coverage gain? Have you ever tested adding these two values? Is there an ablation study?
>
> **A6**: Thank you for the insightful question. Here we compare these two designs:
> - **product** promotes the selection of tokens that are both informative (salient) and non-redundant (bringing new coverage). This multiplicative formulation naturally down-weights tokens that are high in only one of the two dimensions.
> - **additive** could overemphasize either highly salient but redundant tokens or low-saliency tokens with marginal coverage contributions.
>
> We conducted an ablation study to compare the product and addition strategies. As shown in Table D5, the product-based formulation consistently outperforms the additive one across all benchmarks, demonstrating its effectiveness.
>
> **Table D5: Comparison of different combinations of saliency and coverage.**
> |Method|GQA|MMB|MME|POPE|TextVQA|
> |:-:|:-:|:-:|:-:|:-:|:-:|
> |Add | 57.7 | 60.1 | 1647 | 83.4 | 54.9 |
> |Product| 58.3 |  61.7|  1698| 83.9|  56.6|
> |Gain |**+0.6**|**+1.6**|**+51**|**+0.5**|**+1.7**|

---

> > ### Author Response · Authors · 2025-08-04
> > **Appreciation and Invitation for Further Discussion**
> >
> > Dear Reviewer 6ZPM,
> >
> > Thank you once again for your thoughtful review and valuable feedback on our submission.
> > We truly appreciate the time and effort you devoted to evaluating our work.
> >
> > During the rebuttal period, we did our best to address the concerns you raised as thoroughly as possible. We would be very happy to engage in further discussion and clarify any remaining questions or concerns you might have.
> >
> > Best regards,
> >
> > The authors of submission 4718

---

> ### Author Response · Authors · 2025-08-06
> **Second Invitation for Further Discussion**
>
> ### Second Invitation for Further Discussion
>
> Dear Reviewer 6ZPM:
>
> Given the review timeline and policy, we would be grateful for any indication of whether our responses have satisfactorily addressed your concerns, or if further clarification is needed.
>
> We remain committed to improving our manuscript based on your expertise.
>
> Thank you for your time and consideration.
>
> Best regards,
>
> The authors

---

> ### Author Response · Authors · 2025-08-07
> **Gentle Reminder: Reviewer Discussion Participation**
>
> ### Gentle Reminder: Reviewer Discussion Participation
>
> Dear Reviewer 6ZPM,
>
> Thank you again for your time and effort in reviewing our submission.
>
> Just a gentle reminder that the discussion phase will end soon. We truly appreciate your earlier review, and would be grateful for any additional comments or clarifications you might share during this phase.
>
> As noted in the NeurIPS policy, submitting the “Mandatory Acknowledgement” does not replace participation in the discussion. Your engagement would help ensure a more thorough and balanced evaluation.
>
> Please let us know if we can clarify any points from the rebuttal or provide further information.
>
> Best regards,
>
> The Authors of Submission 4718

---

> ### Author Response · Authors · 2025-08-08
> **Clarification Request Regarding Remaining Concerns**
>
> ### Clarification Request Regarding Remaining Concerns
> Dear Reviewer 6ZPM,
>
> Thank you again for your thoughtful review and for the time you've dedicated to evaluating our submission.
>
> As per the NeurIPS 2025 discussion guidelines, reviewers are encouraged to communicate clearly whether their concerns have been addressed:
>
> > **“If authors have resolved your questions, do tell them so.”**
>
> > **“If authors have not resolved your questions, do tell them so too.”**
>
> We have provided detailed responses to the concerns you previously raised. We kindly ask whether there are any remaining issues you feel have not yet been addressed.
>
> Just a gentle reminder that the discussion phase will end soon. We would truly appreciate any additional comments you might have.
>
> Best regards,
>
> The Authors

---

> > ### Comment · Senior_Area_Chairs · 2025-08-08
> >
> > Dear reviewer,
> >
> > Please post a message to let the authors know whether your questions have been resolved. Note that the authors are unable to view your final rating and justification at this time.
> >
> > Thanks,
> > SAC

---

> ### Comment · Area_Chair_sKjX · 2025-08-08
> **Please give your feedbacks**
>
> The author provided the rebuttal. Could you please check if the concerns are fully addressed?
>
> Thanks,
> AC

---

### Official Review · Reviewer_shnr · 2025-06-27

**Clarity:** 2
**Significance:** 3
**Originality:** 2
**Rating:** 4
**Confidence:** 3

**Summary:**

This paper introduces SCOPE, a novel visual token pruning method that combines saliency and coverage to improve semantic completeness. By defining token-coverage gain and integrating saliency scores, SCOPE iteratively selects the most relevant tokens. Extensive experiments on LLaVA models show that SCOPE outperforms previous methods in vision-language tasks.

**Questions:**

Overall, this paper is well done with very few issues, but there are still some problems (see weaknesses). If the author addresses my concerns, I will consider raising the score.

**Ethical Concerns:**

["NO or VERY MINOR ethics concerns only"]

**Final Justification:**

Based on the feedback from other reviewers and the authors' rebuttal, my concerns regarding 'computational and time cost' and 'ablation experiment' have been mainly addressed, and I maintain a positive score.

**Limitations:**

yes

**Quality:**

3

**Strengths And Weaknesses:**

Strengths: the core contribution of this paper is clear: jointly modeling both the saliency and coverage of the selected visual tokens to better preserve semantic completeness. The proposed method achieves this goal, and both visual results and experimental findings support this claim.

Weaknesses:
1. Due to the introduction of an additional module, how much computational and time cost does SCOPE introduce compared to previous methods?
2. Table 3 shows that the increase in the ablation experiment is limited. Could author analyze the reason for this?

---

> ### Author Rebuttal · Authors · 2025-07-31
>
> We sincerely appreciate the reviewer’s recognition of the core contribution of our work, i.e., the joint modeling of saliency and coverage to preserve semantic completeness. We are encouraged that both our visualizations and experimental results were found to effectively support this goal. We address your questions as follows:
>
> > **Q1**: Due to the introduction of an additional module, how much computational and time cost does SCOPE introduce compared to previous methods?
>
> **A1**: Thank you for your question. As shown in Table 4 of the main paper and detailed again in Table C1 below, we analyze the efficiency of our method on LLaVA-NeXT 7B. Although SCOPE introduces slightly higher latency than PDrop, it still achieves a 3.0$\times$ speedup compared to the full-token baseline. Importantly, SCOPE maintains strong performance on the POPE metric (83.0% vs. 86.4%), showing that our token selection strategy preserves semantic completeness.
> In contrast, PDrop exhibits a significant performance drop (53.2% POPE), likely due to its reliance on text-visual saliency-only pruning, which may overlook tokens that are not highly attended but are semantically crucial.
>
> **Table C1: Efficiency analysis of our method on LLaVA-NeXT 7B. The experiments are conducted on a system equipped with $4\times$A100. $\Delta$ denotes the reduction ratio.**
> | Method | Token Number | POPE | Latency (s) | $\Delta$ |
> | :--- | :---: | :---: | :---: | :---: |
> | Vanilla | 2880 | 86.4 | 601.9 | - |
> | PDrop (CVPR'25) | 160 | 53.2 | 184.0 | $3.3\times$ |
> | Ours | 160 | 83.0 | 202.6 | $3.0\times$ |
>
> > **Q2**: Table 3 shows that the increase in the ablation experiment is limited. Could author analyze the reason for this?
>
> **A2**: Thank you for the question. We analyse the ablation experiments as follows:
> 1. **SCOPE achieves significant gains compared to the random and saliency-only baselines**, especially on challenging benchmarks such as POPE and TextVQA. This confirms the importance of informed token selection.
> 2. **The coverage-only variant achieves very strong performance**, validating our key design motivation: ensuring diverse and complete visual semantics is crucial for multimodal reasoning.
> 3. **Saliency provides complementary benefits**. Although its standalone performance is weaker than coverage-only, incorporating it into our full method (SCOPE) consistently leads to further improvements. This suggests that saliency helps refine the focus toward more informative regions that coverage alone might overlook.
>
> In summary, SCOPE effectively integrates both saliency and coverage. The relatively small gain over coverage-only reflects the strength of the coverage, rather than a limitation of our approach. The consistent improvements across all benchmarks highlight the robustness and synergy of the combined design.

---

> > ### Comment · Reviewer_shnr · 2025-08-03
> >
> > Thank you for the rebuttal. My concerns are mainly addressed. I keep the positive rate.

---

> > > ### Author Response · Authors · 2025-08-03
> > >
> > > Thank you for letting us know that your concerns have been mainly addressed and for keeping a positive evaluation.
> > >
> > > We sincerely appreciate your constructive feedback, which has been very helpful in improving our paper. If there are any remaining concerns, we would be glad to clarify or provide further details.

---

### Official Review · Reviewer_Toag · 2025-06-28

**Clarity:** 4
**Significance:** 3
**Originality:** 2
**Rating:** 4
**Confidence:** 5

**Summary:**

The paper proposes SCOPE, a visual token pruning method for VLMs that considers both token saliency and semantic coverage to improve visual token pruning efficiency. Unlike prior approaches that rely solely on attention-based saliency, SCOPE introduces a token-coverage-aware selection mechanism based on pairwise token similarities, ensuring that the selected tokens can cover the most important regions. The method is train-free, integrates seamlessly after the vision encoder and is applicable to a wide range of VLMs. It consistently outperforms state-of-the-art baselines on a range of benchmarks, especially under aggressive token pruning settings.

**Questions:**

1. It lacks a visualization of selected visual tokens for images to help us better understand how the SCOPE score-guided token pruning differs from saliency-guided token pruning. The provided visualization is not an actual image and thus is hard to interpret. For example, you should use a mask to highlight the chosen region of the image.
2. How does the proposed method compare to randomly chosen tokens in terms of performance? Random selection can also enable evenly pruned tokens, thus potentially providing more complete coverage, though this is a very simple and naive heuristic.
3. In Figure 2, the coverage for SCOPE is actually lower than for coverage only, but in the ablation study, SCOPE performs better than coverage only. Can you provide some explanation to help me understand what this means?
4. In Figure 4, when the number of tokens is relatively large (for example, 48), it seems that VisionZip is starting to beat SCOPE. However, in Tables 1 and 2, when retaining 64 / 160 tokens, SCOPE is significantly better than VisionZip. This is inconsistent, as if Figure 4 is true, then VisionZip should be better when the number of retained tokens is 64. How do you explain this?

**Ethical Concerns:**

["NO or VERY MINOR ethics concerns only"]

**Final Justification:**

The rebuttal addresses my concerns, but I still think that there is no large difference between coverage-oriented and saliency-oriented token selection. In other words, I think coverage-oriented token selection is just a supplement to saliency-oriented selection. Considering that the authors provides extensive experiments results to study the impact of coverage-oriented token selection and still has some contributions to the community, I would like to maintain my borderline accept.

**Limitations:**

The authors can provide a breakdown of their method's latency to better understand why the significant token reduction does not translate into substantial latency reduction.

**Paper Formatting Concerns:**

There are no formatting issues.

**Quality:**

4

**Strengths And Weaknesses:**

**Quality**: The idea behind this work is clear and technically sound, as many vision token pruning works focus on pruning based on the attention map (saliency-oriented), and few focus on a new aspect like token similarity (called token coverage in this paper). Through extensive experiments, the authors show that incorporating token coverage into consideration can greatly improve performance, especially under extreme pruning ratios. From motivation to method to experimental results, this submission is clear and of high quality.

**Clarity**: This paper is well written, with clear motivation, detailed formulas explaining their method, and neat experiment tables.

**Significance**: The results are pretty good, especially for extreme pruning ratios, which can inspire other work to push further for more extreme pruning. However, though the reduction is significant, it does not translate into significant latency reduction (Table 4), which can reduce the impact of this work in real-world production. Also, from Figure 6, it seems like the difference between the proposed method and the saliency-based method is that the proposed method chooses tokens more evenly. Randomly choosing tokens could also have the same effect, but the authors do not provide a comparison with random selection to justify any performance difference.

**Originality**: The idea of using set coverage to guide pruning is novel to me, but the idea of token similarity is not novel and can be found in some token merging work, such as [1].

Reference:

[1] Token Merging: Your ViT But Faster

---

> ### Author Rebuttal · Authors · 2025-07-31
>
> Thank you for your recognition of our work and the constructive feedback. We would like to address your concerns individually.
>
> > Q1: The idea of using set coverage to guide pruning is novel to me, but the idea of token similarity is not novel and can be found in some token merging work, such as [1].
>
> **A1**: Thank you for recognizing the novelty of using set coverage in our pruning framework. We acknowledge that token similarity has been widely adopted in previous token compression approaches, including token merging methods such as [1]. However, our work differs fundamentally in both approach and objective:
> - **From token merging to semantic completeness**: We utilize token similarity not for merging redundant tokens, but to introduce a set coverage perspective that explicitly encourages semantic completeness among selected tokens. This represents a paradigm shift from existing methods.
> - **Individual saliency vs. collective semantic completeness**: Current saliency-based pruning methods focus on selecting individually important tokens, which creates a critical issue: semantic incompleteness. These methods optimize for individual token importance without considering the collective semantic representation.
> - **Joint saliency-Coverage optimization**: Our key contribution is to formulate token selection as a joint optimization over saliency and coverage.
>     - **Saliency**: Ensures token-level importance is preserved.
>     - **Coverage**: Guarantees selected tokens span complete semantic regions.
>
> This dual optimization results in superior semantic completeness, which proves particularly critical for downstream reasoning tasks in MLLMs where comprehensive understanding is essential. While we share certain tools (e.g., similarity metrics) with prior works, our principal objective, and mathematical formulation represent a distinct contribution that addresses fundamental limitations in existing approaches.
>
> > **Q2**: It lacks a visualization of selected visual tokens for images to help us better understand how the SCOPE score-guided token pruning differs from saliency-guided token pruning. The provided visualization is not an actual image and thus is hard to interpret. For example, you should use a mask to highlight the chosen region of the image.
>
> **A2**: Thank you for your thoughtful suggestion. We agree that visualizing the selected tokens directly on the original image would greatly improve the interpretability of our method.
> - **Key difference in token selection beavior:** The fundamental distinction between saliency-guided pruning and our SCOPE lines in their selection strategies:
>     - **Saliency-based Pruning** focuses only on the most attended tokens (typically the main object or salient featurs), which may lead to semantic redundancy or incompleteness.
>     - **SCOPE** jointly considers both saliency and semantic coverage, aiming to select tokens that are important and semantically complete to form a more comprehensive representation.
>
> **Visualization Commitment**: We have **already prepared visualizations** that highlight the selected regions on the input images using masks. Due to the rebuttal policy, we cannot include these visualizations in this response. We commit to incorporating these detailed visual comparisons in the camera-ready version to better illustrate the behavior of our method.
>
>
> > **Q3**: How does the proposed method compare to randomly chosen tokens in terms of performance? Random selection can also enable evenly pruned tokens, thus potentially providing more complete coverage, though this is a very simple and naive heuristic.
>
> **A3**: Thank you for the valuable suggestion. The comparison between our method and random token selection is already included in the submission (Table 3), and for clarity, we extract the relevant results in Table B1 below:
>
> **Table B1: Performance comparison between Random selection and our SCOPE.**
> | Method | GQA      | MMB      | MME      | POPE     | TextVQA  |
> | :-: |:-:| :-: | :-: | :-: | :-: |
> | Random | 55.5     | 54.0     | 1556     | 75.2     | 48.4     |
> | Ours   | 58.3 | 61.7 | 1698 | 83.9 | 56.6 |
> | Gain | **+2.8**     |	**+7.7**|	**+142**|	**+8.7**	|**+8.2**|
>
> While random pruning may offer broad token coverage due to its uniformity, it fails to retain critical visual semantics. In contrast, our method jointly considers saliency (to select informative tokens) and semantic coverage, leading to stronger semantic preservation and consistently higher accuracy.
>
>
> > **Q4**: In Figure 2, the coverage for SCOPE is actually lower than for coverage only, but in the ablation study, SCOPE performs better than coverage only. Can you provide some explanation to help me understand what this means?
>
> **A4**: Thank you for your question. The key insight is that **maximizing coverage alone does not necessarily lead to optimal model performance**.
>
> - **Coverage-only strategy** indeed promotes semantic coverage by selecting a wide range of tokens, it often includes tokens with low saliency.
> - **Our SCOPE method formulates token selection as a joint optimization of saliency and coverage**. This means that selected tokens are not only semantically complete, but also highly informative. Though this combined objective may result in a slightly lower raw coverage score compared to the coverage-only strategy, the selected tokens better represent the essential semantics of the image, leading to improved model performance.
>
> For better understanding, we provide an example to illustrate the difference of selected tokens among different strategies. As we cannot upload figures, we will briefly describe the visualization of our token pruning strategies on an image of a cat and a banana.
> 1. **Saliency-only**: Selected tokens are mainly concentrated on the cat and banana, demonstrating object-level focus by pruning the background.
> 2. **Coverage-only**: Tokens are spread across the image, preserving global context but potentially missing important object details.
> 3. **Our SCOPE**: High token density is maintained on the cat and banana, while a sparse set of tokens is strategically kept for the background. This captures critical object features without discarding essential scene context.
>
> We have **prepared visualizations to support this observation** and will include them in the final version of the paper.
>
>
> > **Q5**: In Figure 4, when the number of tokens is relatively large (for example, 48), it seems that VisionZip is starting to beat SCOPE. However, in Tables 1 and 2, when retaining 64 / 160 tokens, SCOPE is significantly better than VisionZip. This is inconsistent, as if Figure 4 is true, then VisionZip should be better when the number of retained tokens is 64. How do you explain this?
>
> **A5**: Thank you for your careful observation. We have re-checked the results and confirmed their correctness. The apparent inconsistency between Figure 4 and Tables 1 and 2 mainly arises from performance fluctuations that commonly occur when the number of retained tokens is small.
>
> In low-token regimes (e.g., 32 or 48 tokens), token pruning methods are more sensitive to which specific tokens are preserved, and small differences in selection may lead to noticeable performance variation. For example, as shown in Figure 4 on the SQA dataset, retaining 32 tokens leads to better performance than 48 tokens.
>
> While performance may fluctuate at certain token counts, SCOPE generally **achieves better results than VisionZip across a wide range of benchmarks and token budgets**. We will make this clearer in the final version to ensure clarity.
>
> > **Q6**: The authors can provide a breakdown of their method's latency to better understand why the significant token reduction does not translate into substantial latency reduction.
>
> **A6**: Thank you for the helpful suggestion. To better understand the relationship between token reduction and actual latency improvement, we provide a detailed latency breakdown based on LLaVA-NeXT 7B, shown in Table B2.
>
> **Table B2: Efficient Analysis on LLaVA-NeXT 7B. The experiments are conducted on 4\*A100 on POPE.**
> |   Method  | #Token | Total(s) | Speedup | Pre-LLM(s) | Speedup | LLM Prefilling (s) | Speedup |LLM Decoding| Speedup|
> |:--:|:--:|:--:|:--:|:--:|:--:|:--:|:--:|:--:|:--:|
> | LLaVA-NeXT | 2880  |  601.9  | 1.0$\times$ |    58.9    | 1.0$\times$|  487.3 | 1.0$\times$ |55.7|1.0$\times$|
> |SCOPE | 160|202.6| 3$\times$ | 75.4 | 0.8 $\times$| 79.1 | 6.2 $\times$ |47.7|1.2$\times$|
>
> As shown above, SCOPE significantly reduces the number of visual tokens (from 2880 to 160), resulting in a 6.2$\times$ speedup in the LLM prefilling stage, which is the most time-consuming part of the pipeline. However, this reduction does not linearly translate to overall latency reduction, primarily because:
> - The Pre-LLM stage (vision encoding and token pruning) becomes more dominant after pruning, as its computation is less affected by the token count.
> - The text tokens themselves also contribute non-negligible computation, and their processing cost remains constant regardless of visual token pruning.

---

> > ### Comment · Reviewer_Toag · 2025-08-02
> >
> > Thank you for the rebuttal. My concerns are addressed.

---

> > > ### Author Response · Authors · 2025-08-03
> > >
> > > Thank you for your insightful comments and for acknowledging that we have addressed your concerns.
> > >
> > > Your expert feedback was invaluable in helping us strengthen the paper. We would be grateful if you would consider these improvements in your final evaluation.

---

### Official Review · Reviewer_h3Fi · 2025-07-03

**Clarity:** 3
**Significance:** 2
**Originality:** 3
**Rating:** 4
**Confidence:** 4

**Summary:**

This paper tackles the computational overhead in Multimodal Large Language Models (MLLMs) caused by excessive visual tokens. The authors argue that existing saliency-based pruning methods compromise semantic "coverage." To address this, they propose SCOPE (Saliency-Coverage Oriented token Pruning), a novel, training-free framework that jointly optimizes for both saliency and coverage. SCOPE uses a greedy algorithm to iteratively select tokens that maximize a score combining intrinsic saliency and marginal coverage gain. Extensive experiments on LLaVA-1.5 and LLaVA-Next show that SCOPE consistently outperforms prior methods, especially under high compression rates.

**Questions:**

N/A

**Ethical Concerns:**

["NO or VERY MINOR ethics concerns only"]

**Final Justification:**

I have reviewed the authors’ response as well as their discussion with the other reviewers. I would like to maintain my recommendation to accept the paper.

**Quality:**

3

**Strengths And Weaknesses:**

**Strengths**

The greedy algorithm, based on maximizing a marginal gain (Eq. 7 & 8), is a principled approach reminiscent of submodular optimization, making it well-suited for this diverse subset selection problem.

The SCOPE score (Eq. 9) provides a simple yet effective mechanism to balance the competing objectives of saliency and coverage.

Being training-free, the method is highly practical and can be easily integrated into existing MLLM inference pipelines.

**Weaknesses**

While the ultimate application of the paper is in MLLMs, the core technical contribution, aka the SCOPE algorithm, is a module that operates purely on the vision side. It completes all pruning operations before the visual tokens are fed into the language model. Therefore, confining the experimental comparison solely to other pruning methods designed specifically for MLLMs presents an incomplete evaluation.

A more rigorous evaluation should have benchmarked SCOPE against established token pruning or merging methods designed for ViTs. These methods are the precursors and critical baselines for addressing the same fundamental problem: how to efficiently process a large number of visual tokens.

If SCOPE cannot demonstrate a clear advantage over these mature vision-side pruning methods in end-to-end MLLM tasks, its novelty and practical value are significantly diminished. In other words, a key question remains unanswered: Do we need a new, MLLM-specific visual pruning method like SCOPE, or could we simply "plug-and-play" existing state-of-the-art vision pruning methods into MLLMs and achieve comparable or even better results?

---

> ### Author Rebuttal · Authors · 2025-07-31
>
> We sincerely appreciate the reviewer’s valuable and constructive feedback. Below, we address your questions as follows:
>
> > **Q1**: Comparison with methods of vision token compression for ViTs.
>
> **A1**: Thank you for the suggestion. We agree that established ViT-based token pruning methods are important baselines. Following your suggestion, we compare our method with a representative ViT-based token pruning approach, Token Merging (ToMe) [1], which reduces token redundancy while preserving visual feature quality. The results on LLaVA-1.5 7B are presented in Table A1.
>
> **Table A1: Results of SCOPE and ViT token pruning method.**
> |Method|#Token|GQA|MMB|MME|POPE|SQA|TextVQA|SEED|MMVet|Avg.|
> |:-:|:-:|:-:|:-:|:-:|:-:|:-:|:-:|:-:|:-:|:-|
> |Vanilla|576|61.9|64.7| 1862 | 85.9| 69.5|58.2|58.6| 31.1| 100% |
> |ToMe | 192 | 54.3 | 60.5 | 1563 | 72.4 | 65.2 | 52.1 | 53.1 | 27.9| 88.9%|
> |**Ours** | 192 | 60.1 | 63.6 | 1804 | 86.4 |  68.8 | 57.7 | 58.7 | 32.5 | 99.5% (**+10.6%**)|
> |ToMe | 128 | 52.4 | 53.3 | 1343 | 62.8 | 59.6 | 49.1 | 50.9 | 27.2 | 81.9%|
> |**Ours** | 128 | 59.7 | 62.5 | 1776 | 86.1 | 68.4 |  57.2 |  57.8 |  31.4 | 98.1% (**+16.2%**) |
> |ToMe | 64 | 48.6| 43.7|1138| 52.5| 50.0|45.3|44.0|24.1| 71.1% |
> |**Ours** |64| 58.3 |  61.7|  1698|  83.9|  68.6|  56.6|  56.3|  30.4| 96.0% (**+24.9%**) |
>
> From Table A1, we can see that our SCOPE consistently outperforms ToMe under the same token budget. These results highlight the importance of developing vision-side token pruning strategies specifically tailored for MLLMs, and validate the effectiveness of our proposed method. This trend is also observed in prior efficient MLLM designs such as VisionZip, which adopt post-ViT pruning strategies.
>
> The key reasons behind the performance differences are:
> - **Different semantic requirements:**
>     - ViT-based pruning methods aim to retain discriminative features for classification tasks, which primarily require category-level information.
>     - MLLMs demand more comprehensive and fine-grained semantic representations to support image understanding and reasoning.
> - **Different pruning stages:**
>     - **ViT-based methods prune tokens at intermediate layers to reduce computation within the ViT backbone**. In ViTs, as layers go deeper, the relationships among tokens evolve. Shallow layers focus primarily on local spatial structures, while deeper layers capture more global and semantic dependencies. Therefore, pruning too early may disrupt the development of critical token interactions before they are fully established.
>     - **Vision-side token pruning for MLLMs, in contrast, is applied after the full forward pass of the ViT**. This allows pruning decisions to be based on the final visual representations, which already encode rich semantic information. Such a design better aligns with the needs of MLLMs, enabling more informed token selection based on comprehensive embeddings.
>
> We will include this comparison and discussion in the final version to provide a more comprehensive evaluation.
>
> We thank the reviewer again for the insightful comment. Although existing ViT-based pruning methods are not directly compatible with MLLMs, we believe it is a promising future direction to design ViT-side token merging/pruning that is also aware of multimodal semantic needs.
>
> [1] Token merging: Your vit but faster, ICLR 2023.

---

> ### Author Response · Authors · 2025-08-06
> **Second Invitation for Further Discussion**
>
> ### Second Invitation for Further Discussion
>
> Dear Reviewer h3Fi:
>
> Given the review timeline and policy, we would be grateful for any indication of whether our responses have satisfactorily addressed your concerns, or if further clarification is needed.
>
> We remain committed to improving our manuscript based on your expertise.
>
> Thank you for your time and consideration.
>
> Best regards,
>
> The authors

---

> ### Author Response · Authors · 2025-08-07
> **Gentle Reminder: Reviewer Discussion Participation**
>
> ### Gentle Reminder: Reviewer Discussion Participation
>
> Dear Reviewer h3Fi,
>
> Thank you again for your time and effort in reviewing our submission.
>
> Just a gentle reminder that the discussion phase will end soon. We truly appreciate your earlier review, and would be grateful for any additional comments or clarifications you might share during this phase.
>
> As noted in the NeurIPS policy, submitting the “Mandatory Acknowledgement” does not replace participation in the discussion. Your engagement would help ensure a more thorough and balanced evaluation.
>
> Please let us know if we can clarify any points from the rebuttal or provide further information.
>
> Best regards,
>
> The Authors of Submission 4718

---

> > ### Comment · Reviewer_h3Fi · 2025-08-08
> >
> > My concerns are addressed.

---

> ### Author Response · Authors · 2025-08-08
> **Clarification Request Regarding Remaining Concerns**
>
> ### Clarification Request Regarding Remaining Concerns
> Dear Reviewer h3Fi,
>
> Thank you again for your thoughtful review and for the time you've dedicated to evaluating our submission.
>
> As per the NeurIPS 2025 discussion guidelines, reviewers are encouraged to communicate clearly whether their concerns have been addressed:
>
> > **“If authors have resolved your questions, do tell them so.”**
>
> > **“If authors have not resolved your questions, do tell them so too.”**
>
> We have provided detailed responses to the concerns you previously raised. We kindly ask whether there are any remaining issues you feel have not yet been addressed.
>
> Just a gentle reminder that the discussion phase will end soon. We would truly appreciate any additional comments you might have.
>
> Best regards,
>
> The Authors

---

> > ### Comment · Senior_Area_Chairs · 2025-08-08
> >
> > Dear reviewer,
> >
> > Please post a message to let the authors know whether your questions have been resolved. Note that the authors are unable to view your final rating and justification at this time.
> >
> > Thanks,
> > SAC

---

> ### Comment · Area_Chair_sKjX · 2025-08-08
> **Please give your feedbacks**
>
> The author provided the response. Could you please check if the concerns are fully addressed?
>
> Thanks,
> AC

---

### Decision · Program_Chairs · 2025-09-17

**Decision:**

Accept (poster)

**Comment:**

This paper received overall positive reviews finally. Reviewers acknowledged that it introduces an efficient token pruning approach that is practical and easily integrated into existing MLLM inference pipelines. The rebuttal effectively addressed initial concerns, including incomplete comparisons and computational and time costs, clarifying the paper to an acceptance-worthy level. However, some reviewers still expressed concerns about the limited scope and overall contribution to the community. The AC concurs with the majority and considers this a solid contribution, recommending acceptance as a poster. It is advised that the clarifications and supporting evidence from the rebuttal be incorporated into the camera-ready version. We look forward to seeing this work presented at NeurIPS.